# The catalytic subunit of DNA polymerase δ inhibits γTuRC activity and regulates Golgi-derived microtubules

Yuehong Shen [1], Pengfei Liu [1], Taolue Jiang [1], Yu Hu[1], Franco K.C. Au [1] & Robert Z. Qi [1]

γ-Tubulin ring complexes (γTuRCs) initiate microtubule growth and mediate microtubule attachment at microtubule-organizing centers, such as centrosomes and the Golgi complex. However, the mechanisms that control γTuRC-mediated microtubule nucleation have remained mostly unknown. Here, we show that the DNA polymerase δ catalytic subunit (PolD1) binds directly to γTuRCs and potently inhibits γTuRC-mediated microtubule nucleation. Whereas PolD1 depletion through RNA interference does not influence centrosome-based microtubule growth, the depletion augments microtubule nucleation at the Golgi complex. Conversely, PolD1 overexpression inhibits Golgi-based microtubule nucleation. Golgi-derived microtubules are required for the assembly and maintenance of the proper Golgi structure, and we found that alteration of PolD1 levels affects Golgi structural organization. Moreover, suppression of PolD1 expression impairs Golgi reassembly after nocodazole-induced disassembly and causes defects in Golgi reorientation and directional cell migration. Collectively, these results reveal a mechanism that controls noncentrosomal γTuRC activity and regulates the organization of Golgi-derived microtubules.

[1] Division of Life Science and State Key Laboratory of Molecular Neuroscience, The Hong Kong University of Science and Technology, Clear Water Bay, Kowloon, Hong Kong. Correspondence and requests for materials should be addressed to R.Z.Q. (email: qirz@ust.hk)

The microtubule cytoskeleton plays a major role in the organization and distribution of organelles in animal cells. In interphase cells, microtubule arrays are generally focused at the centrosome and also at the Golgi complex, a membranous organelle that typically surrounds the centrosome. Whereas a radially symmetrical array of microtubules emanates from the centrosomes, substantial amounts of the Golgi-associated microtubules are arranged in asymmetric patterns[1, 2]. The Golgi-associated microtubules participate in numerous activities, including Golgi ribbon assembly and structural maintenance and cell polarization and directional migration[2–7].

The organization of cellular microtubules by the centrosome and the Golgi complex requires γ-tubulin, a highly conserved protein that plays a key role in the nucleation and then minus-end capping of microtubules[8–11]. γ-Tubulin exists in two complexes: the γ-tubulin small complex (γTuSC) and the γ-tubulin ring complex (γTuRC). Whereas the γTuSC is a tetramer consisting of two γ-tubulins and one molecule each of GCP2 and GCP3, the γTuRC is a macromolecular structure comprising several γTuSCs and other proteins, such as GCP4, GCP5, and GCP6. In each γTuRC, γTuSCs and GCP 4–6, which are the core components, are arranged into a ring-shaped structure; the closure of the ring allows the assembled structure to act as a template for the initiation of microtubule growth[12–15]. Furthermore, γTuRCs are the principal nucleators of cellular microtubules and are required for the microtubule-organizing function of all recognized microtubule-organizing organelles and sites[9–11]. The microtubule-nucleating function of γTuRCs is under a stringent spatiotemporal control by unknown mechanisms. For example, although most of the cellular γ-tubulin exists in a noncentrosomal cytosolic pool and the majority of the cytosolic γ-tubulin is present in γ-tubulin complexes, the cytosolic complexes display very low or almost no microtubule-nucleating activity[16, 17]. In mammalian cells, γTuRCs are recruited to microtubule-organizing centers, where the complexes mediate microtubule nucleation and anchoring of the microtubules.

Several proteins have been found to interact with γTuRCs and participate in γTuRC recruitment to centrosomes and the Golgi complex. One of these proteins is CDK5RAP2, a centrosomal scaffold protein that interacts with γTuRCs through a short sequence that is conserved in γ-tubulin complex-tethering proteins in organisms ranging from yeast to mammals[17, 18]. The binding of this CDK5RAP2 domain stimulates the microtubule-nucleating activity of γTuRCs, and therefore the domain is called the γTuRC-mediated nucleation activator (γTuNA)[17]. By exploiting the specific interaction that occurs with the γTuNA, we established a method of capturing γTuRCs from HEK293T cell cultures[17, 19], and, in this study, we identified the DNA polymerase δ (Pol δ) catalytic subunit (PolD1) as one of the captured proteins. Our data show that PolD1 acts as an inhibitor of γTuRCs, and further that PolD1 controls γTuRC-mediated microtubule nucleation at the Golgi complex and, consequently, regulates several events that require Golgi-derived microtubules. These results not only reveal a mechanism for controlling cytoplasmic γTuRC activities, but also demonstrate a previously unrecognized function of PolD1, a major enzyme in DNA replication and repair.

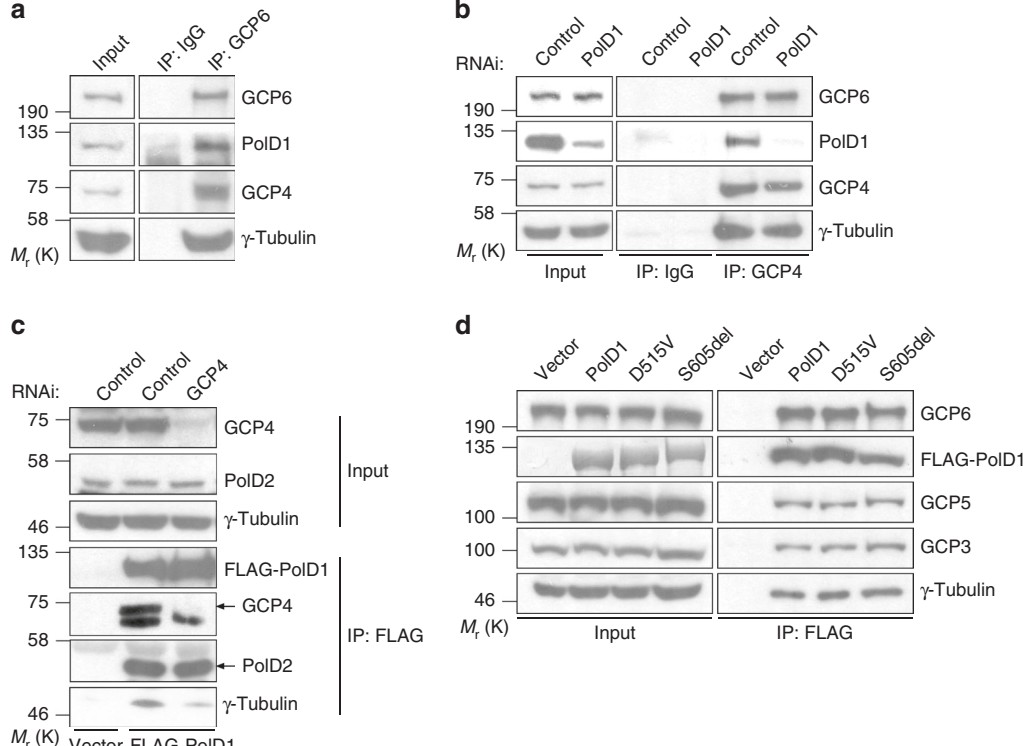

**Fig. 1** PolD1 associates with γTuRCs. **a** Anti-GCP6 immunoprecipitation was performed using HEK293T cell extracts; normal rabbit IgG was used in the control immunoprecipitation. The immunoprecipitates were immunoblotted with the indicated antibodies. **b** HEK293T cells were transfected with control or *pold1*-targeting siRNAs, and then immunoprecipitation was performed using normal rabbit IgG or anti-GCP4 antibody. **c** FLAG-PolD1 was expressed in cells that were transfected with a GCP4-specific siRNA (or a control siRNA), and then anti-FLAG immunoprecipitation was performed. The immunoprecipitates and the lysate inputs were immunoblotted with the indicated antibodies. FLAG-PolD1 was detected through anti-FLAG immunoblotting. **d** Extracts of HEK293T cells ectopically expressing PolD1 or its mutants (D515V and S605del) were used for anti-FLAG immunoprecipitation

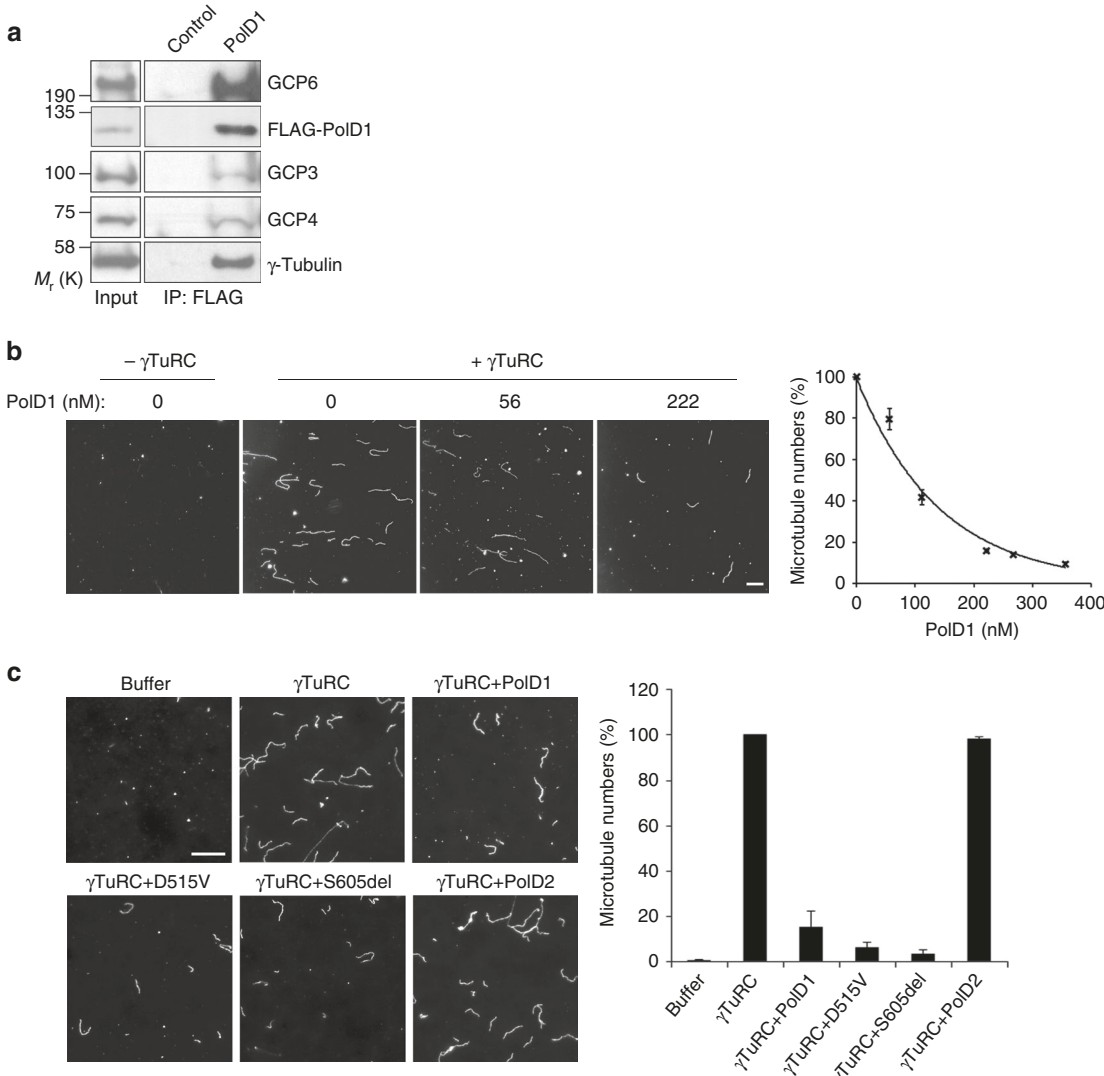

**Fig. 2** PolD1 inhibits γTuRC-induced microtubule nucleation in vitro. **a** The recombinant protein FLAG-PolD1 was tested for binding with purified γTuRCs. FLAG-PolD1 was absent in the control. After anti-FLAG immunoprecipitation, the input γTuRCs and the immunoprecipitates were immunoblotted. **b** Purified γTuRCs reconstituted with the γTuRC stimulator CDK5RAP2(51–200) were used in the microtubule-nucleating assay together with PolD1 at various concentrations. Representative microscopic fields of polymerized microtubules are shown. The data are presented as means ± s.e.m. of three independent experiments. **c** Purified γTuRCs were used for microtubule nucleation in the absence of CDK5RAP2(51–200). PolD1 wild-type and mutants (D515V and S605del) and PolD2 were added at an excessive amount (356 nM). Shown are representative images of polymerized microtubules, and the data are presented as means ± s.d. of three independent experiments. *Scale bars*, 10 μm

## Results

**PolD1 associates with γTuRCs.** In the isolated γTuRCs, we detected PolD1, in addition to γ-tubulin and GCP 2–6, by using mass spectrometry (Supplementary Table 1). Pol δ is a major DNA replicative polymerase and it is also involved in DNA repair and recombination[20–23]. Pol δ consists of the catalytic subunit PolD1 (p125) and three accessory subunits, PolD2 (p50), PolD3 (p68), and PolD4 (p12). Among these subunits, PolD1 is highly conserved among eukaryotes and contains two functional domains: an exonuclease domain near the amino terminus that catalyzes 3′ → 5′ exonucleolytic proofreading, and the subsequent polymerase domain that catalyzes DNA synthesis[20–23].

To verify the association of PolD1 with γTuRCs, we performed anti-GCP6 immunoprecipitation. PolD1 was specifically detected in anti-GCP6 immunoprecipitates, which also contained other γTuRC subunits (Fig. 1a). Similarly, anti-GCP4 immunoprecipitation coprecipitated PolD1 (Fig. 1b). However, RNAi-mediated depletion of PolD1 did not affect the coimmunoprecipitation of

γTuRC subunits such as γ-tubulin and GCP6 with GCP4 (Fig. 1b), which indicates that PolD1 is dispensable for γTuRC assembly. In a reciprocal experiment, γTuRC subunits such as γ-tubulin and GCP 3–6 were coimmunoprecipitated with transiently expressed PolD1 (Fig. 1c, d). To test whether the association of PolD1 with γTuRC subunits requires the presence of intact γTuRCs, we disrupted cellular γTuRC assembly by suppressing the expression of GCP4[17], and then immunoprecipitated the PolD1 that was ectopically expressed in the cells. GCP4 depletion drastically reduced (by ~75%) the coprecipitation of γ-tubulin with PolD1 (Fig. 1c). These data showed that PolD1 interacts primarily with the γ-tubulin that is present in γTuRCs. PolD2, a Pol δ subunit that serves as a scaffold and binds directly to PolD1[24], was coimmunoprecipitated with PolD1 both in the presence and absence of γTuRCs (Fig. 1c).

To determine whether the DNA polymerase and exonuclease activities of PolD1 are involved in the protein's association with γTuRCs, we tested the γTuRC interaction of two PolD1 mutants,

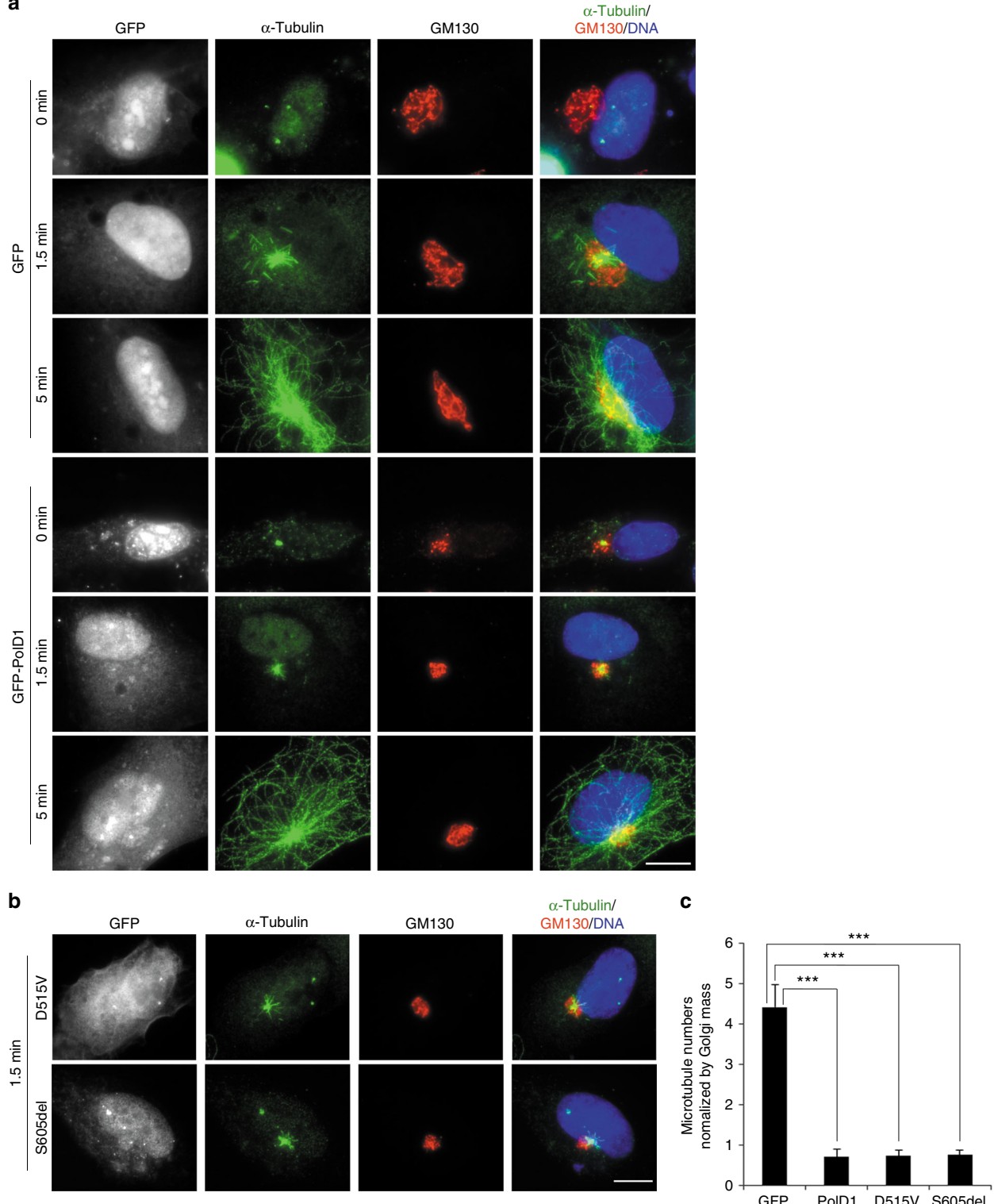

**Fig. 3** PolD1 overexpression inhibits Golgi-associated microtubule regrowth. **a**, **b** RPE1 cells were transfected with GFP or GFP-PolD1 constructs and then subject to cold-induced microtubule depolymerization, after which microtubule regrowth was performed and examined through immunostaining. Hoechst 33258 was used to stain DNA. In the cells transfected with the PolD1 mutants D515V and S605del, microtubule regrowth was for 1.5 min (**b**). The images shown represent the phenotypes identified from three experiments (100 cells were analyzed per sample, and cells expressing GFP, GFP-PolD1, or PolD1 mutants at similar levels were selected for analysis). *Scale bars*, 10 μm. **c** Golgi-associated microtubules at 1.5 min of regrowth were quantified. The quantified data were normalized by the Golgi mass, and are presented as means ± s.d. from three independent experiments; ***$p < 0.001$, two-tailed, unpaired student's *t*-test

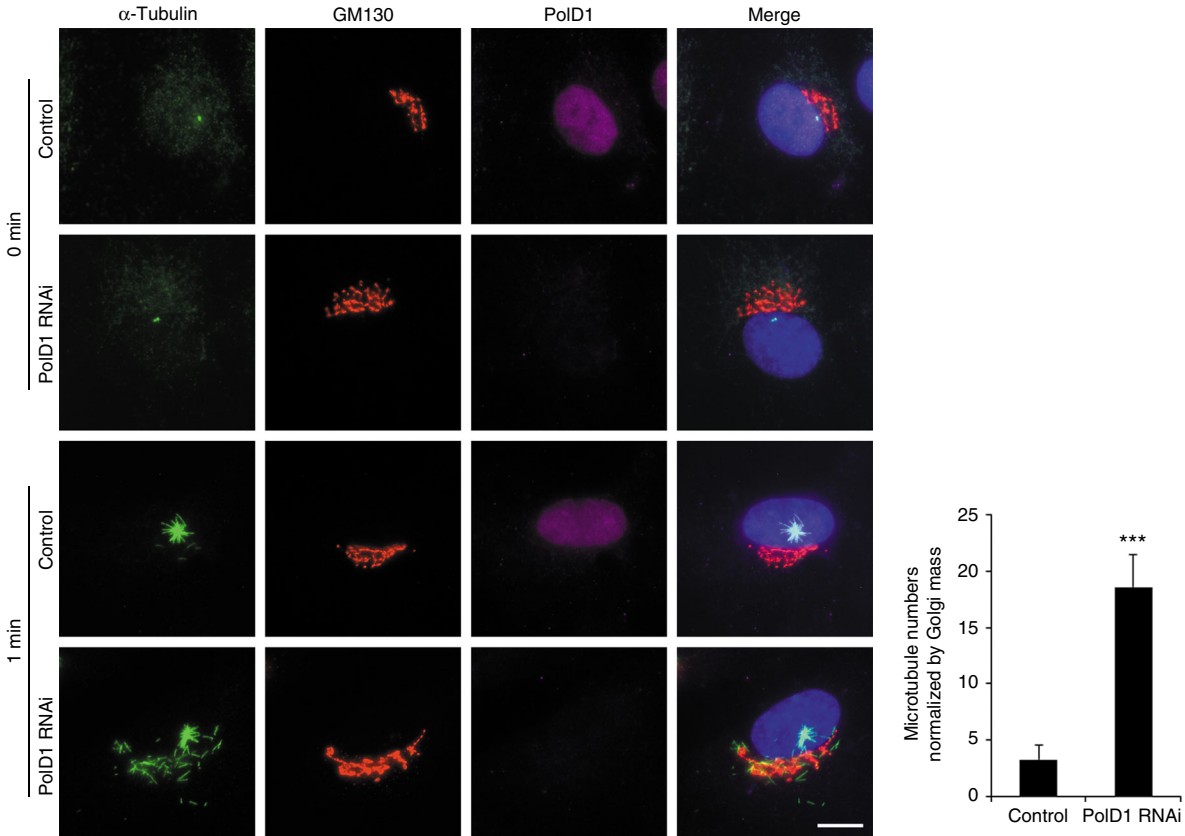

**Fig. 4** PolD1 depletion promotes Golgi-associated microtubule nucleation. A microtubule regrowth assay was performed on RPE1 cells transfected with control or PolD1 siRNAs. The cells were then stained for microtubules (anti-α-tubulin) and GM130. The images shown are representative of at least three experiments (100 cells analyzed per condition). *Scale bar*, 10 μm. Golgi-associated microtubules at 1 min of regrowth were counted and normalized by the Golgi mass. The data are presented as means ± s.d. and are representative of three independent experiments; ***$p < 0.001$, two-tailed, unpaired student's *t*-test

the polymerase-deficient mutant S605del (an in-frame deletion of S605 at the polymerase active site) and the exonuclease-deficient mutant D515V[25, 26]. In immunoprecipitation assays, both mutants showed similar γTuRC-binding activity as wild-type PolD1 (Fig. 1d). Therefore, PolD1 associates with γTuRCs in a manner independent of its exonuclease and polymerase activities.

**PolD1 is a potent inhibitor of γTuRCs.** To investigate the function of the PolD1 that is associated with γTuRCs, we conducted in vitro reconstitution assays. For these assays, we isolated γTuRCs by exploiting their specific interaction with the γTuRC-binding domain of CDK5RAP2[17–19]. After tandem affinity purification through the ectopic tags of the CDK5RAP2 domain and subsequent desalting, γTuRCs were obtained at high purity (Supplementary Fig. 1a). Similarly, PolD1 was ectopically expressed in HEK293T cells and then isolated under a stringent condition (washing with a buffer containing 1 M NaCl and 0.1% IGEPAL CA-630) to remove all bound proteins, including γTuRCs (Supplementary Fig. 1b, c). Subsequently, we confirmed the physical interaction between γTuRCs and PolD1 by testing the binding of the purified proteins. After the proteins were incubated together, γTuRCs were readily coimmunoprecipitated with PolD1 (Fig. 2a).

We tested in vitro the effect of PolD1 on γTuRC-dependent microtubule nucleation. Purified γTuRCs exhibit a low level of microtubule-nucleating activity, and this activity is stimulated upon binding to the γTuNA of CDK5RAP2[17]. We measured the microtubule-nucleating activity of the γTuRCs reconstituted with this CDK5RAP2 domain in the presence of PolD1 added

at various concentrations. Notably, γTuRC-induced microtubule nucleation was inhibited by PolD1 in a dose-dependent manner, with the IC$_{50}$ being 98 nM and ~90% of the activity being inhibited upon adding 356 nM PolD1 (Fig. 2b). Next, we assayed the PolD1 effect on γTuRC-dependent microtubule nucleation in the absence of the CDK5RAP2 protein. Here, in addition to wild-type PolD1, we tested the D515V and S605del mutants and another Pol δ subunit, PolD2. The PolD1 mutants and PolD2 were expressed in HEK293T cells and isolated in the same manner as wild-type PolD1 (Supplementary Fig. 1b, c). The addition of the PolD1 proteins strongly inhibited γTuRC-mediated nucleation, and wild-type PolD1 and the mutants showed similar inhibitory activities (Fig. 2c). By contrast, the nucleation was unaffected by PolD2 addition (Fig. 2c). These results indicate that the microtubule-nucleating activity of γTuRCs is inhibited specifically by PolD1, in a manner independent of the PolD1 DNA polymerase and exonuclease activities.

**PolD1 controls microtubule growth at the Golgi complex.** We overexpressed PolD1 in cultured cells and tested its effect on cellular microtubule nucleation. Here, we used hTERT-RPE1 (RPE1) cells because the centrosome and the Golgi complex organize the two most prominent arrays of microtubules in these cells[2]. In PolD1-overexpressing cells, the centrosomal localization of two γTuRC subunits, γ-tubulin and GCP6, was not discernibly affected (Supplementary Fig. 2a, Fig. 6b). Next, we examined microtubule regrowth after cold-induced depolymerization. At 1.5 min of regrowth, a centrosome-based microtubule aster and

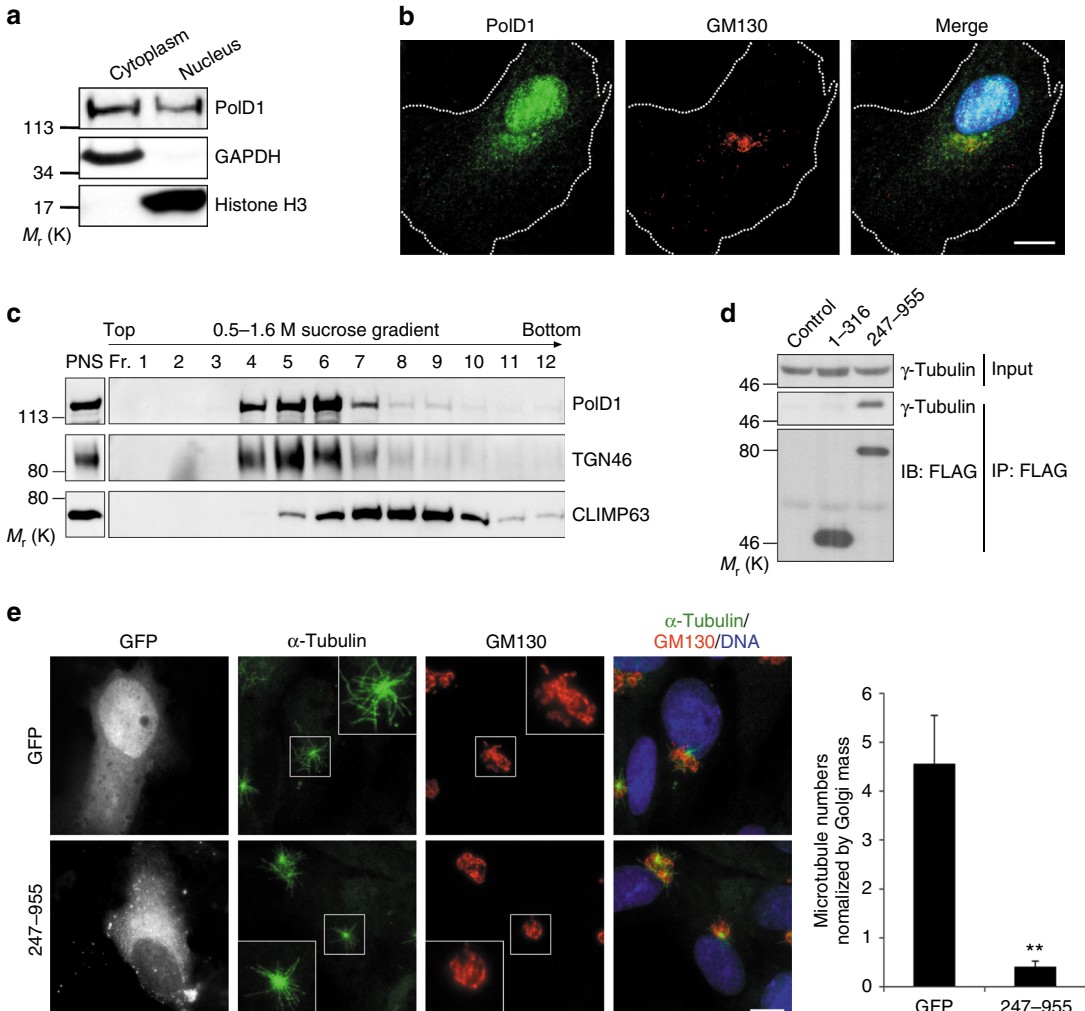

**Fig. 5** Subcellular distribution of PolD1. **a** RPE1 cell lysates were fractionated into cytoplasmic (Cytoplasm) and nuclear (Nucleus) fractions, and an equal proportion of each fraction was immunoblotted. The cytoplasmic protein GAPDH and the nuclear protein histone H3 were probed as fractionation controls. **b** RPE1 cells were double-stained for PolD1 and GM130. The images show a representative cell of at least three experiments; the *white dashed lines* indicate the cell boundary. The PolD1 intensity at the Golgi is ~ 22% of that in the nucleus. **c** The PNS from RPE1 cells was fractionated by equilibrium centrifugation over a sucrose gradient (0.5–1.6 M). Membranes were then pelleted, and each fraction (50%) and the PNS (4%) were analyzed. **d** The PolD1 fragments 1–316 and 247–955 were FLAG-tagged and ectopically expressed in HEK293T cells, and the extracts were used for anti-FLAG immunoprecipitation. The lysate inputs and the immunoprecipitates were analyzed by immunoblotting (*IB*). Control, FLAG vector. **e** A microtubule regrowth assay was performed on RPE1 cells transfected with GFP or GFP fused with the PolD1 fragment 247–955. The cells were then stained for microtubules (anti-α-tubulin) and GM130. The images shown are representative of at least three experiments (20 cells expressing GFP or GFP-tagged 247–955 at similar levels were analyzed at 1.5 min of regrowth). The numbers of Golgi-associated microtubules were determined and normalized by the Golgi mass. The data are presented as means ± s.d. and are representative of three independent experiments; ***$p < 0.001$, two-tailed, unpaired student's *t*-test. *Scale bars*, 10 μm

multiple Golgi-derived short microtubules were observed in control cells (Fig. 3a, c). By comparison, in the PolD1-overexpressing cells, microtubules formed a smaller aster at centrosomes and exhibited almost no Golgi-associated regrowth (Fig. 3a, c). After a longer period (5 min) of regrowth, the PolD1-overexpressing cells displayed a radial array of centrosomal microtubules, but lacked Golgi-derived microtubules (Fig. 3a). Several proteins, such as AKAP450, CDK5RAP2, and CLASPs, participate in microtubule nucleation and growth at the Golgi complex[2–4, 6]. We found that the Golgi localization of AKAP450, CDK5RAP2, and CLASPs was unaffected by the PolD1 overexpression (Supplementary Fig. 2b–d). We also tested the two PolD1 mutants, D515V and S605del, in the regrowth assay. Similar to the wild-type protein, both mutants potently inhibited microtubule regrowth at the Golgi complex and only slightly reduced centrosome-based regrowth (Fig. 3b, c).

These results indicate that PolD1 inhibits Golgi-associated microtubule nucleation by a mechanism that is independent of its exonuclease and polymerase activities.

To further evaluate the effect of PolD1 on cellular microtubule nucleation, we effectively suppressed PolD1 expression by transfecting cells with siRNA duplexes (Supplementary Fig. 3a) and then performed the microtubule regrowth assay. PolD1 depletion caused only a slight increase (~10%) of S-phase cells, and the depletion did not alter the centrosomal content of either γ-tubulin or the γTuRC stimulator CDK5RAP2 (Supplementary Fig. 3b). However, after a short period of regrowth, when few Golgi-derived microtubules were visible in the control cells, robust short microtubules were detected at the Golgi complex of PolD1-depleted cells: the PolD1-depleted cells contained nearly sixfold more Golgi-derived microtubules than the control cells (Fig. 4). By contrast, PolD1 depletion did not markedly

affect centrosome-based microtubule nucleation (Fig. 4). AKAP450 is known to be essential for γTuRC recruitment to the Golgi complex[3, 4], and, accordingly, the depletion here of AKAP450 in addition to PolD1 resulted in an inhibition of the Golgi-based microtubule nucleation (Supplementary Fig. 3c), similar to what was previously observed after knockdown of AKAP450 alone[3, 4]. Therefore, PolD1 depletion substantially augments γTuRC-dependent nucleation of microtubules at the Golgi complex. Furthermore, when we expressed PolD1 or its D515V or S605del mutant in the cells that were depleted of

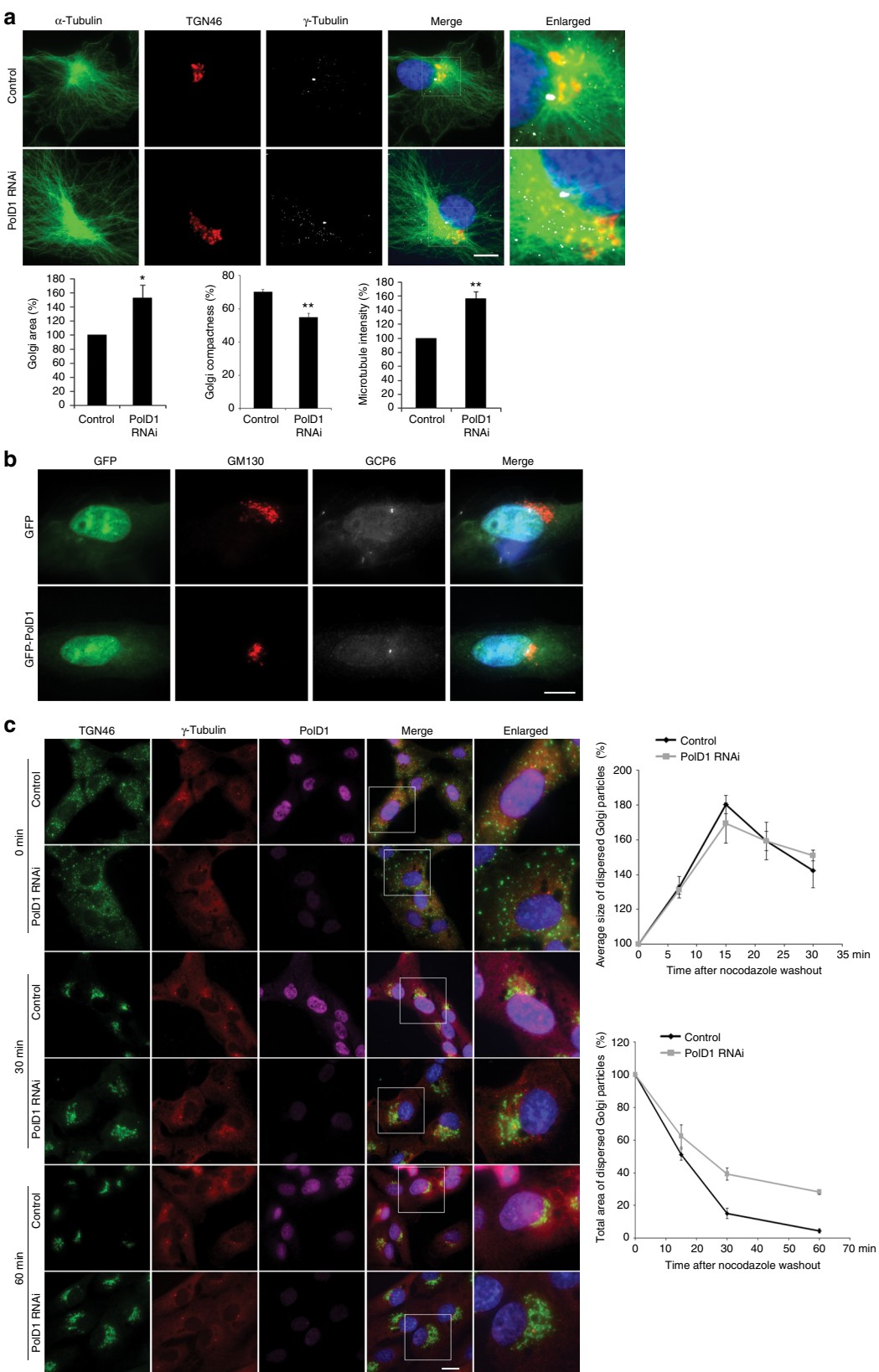

endogenous PolD1, Golgi-based microtubule nucleation was diminished considerably (Supplementary Fig. 4). These results indicate that the augmentation of microtubule nucleation observed after transfection of *pold1*-targeting siRNAs was specifically due to the depletion of PolD1.

To enhance our understanding of PolD1 action, we assessed PolD1 nuclear and cytoplasmic distribution and localization at the Golgi complex. Subcellular fractionation and subsequent immunoblotting revealed that the nucleus and the cytoplasm contained ~37 and ~63% of PolD1, respectively (Fig. 5a). Similarly, transiently expressed GFP-PolD1 was distributed almost equally between the nuclear and cytoplasmic fractions (Supplementary Fig. 5). The nuclear localization of GFP-PolD1 was specific, because GFP alone was present almost exclusively in the cytoplasmic fraction (Supplementary Fig. 5). We next analyzed whether PolD1 localizes at the Golgi. To visualize cytoplasmic PolD1, we adopted a method that was previously described for immunostaining Golgi-associated γ-tubulin:[4] we extracted cells with a saponin-containing buffer to reduce cytosolic staining and then used tandem secondary antibodies to enhance staining sensitivity. When this method was used, PolD1 was clearly observed at the Golgi complex (Fig. 5b). Moreover, the results of complementary biochemical analyses validated the association of PolD1 with Golgi membranes: when the post-nuclear supernatant (PNS) of RPE1 cells was fractionated by means of centrifugation through a sucrose gradient, PolD1 co-fractionated with the Golgi protein TGN46 but showed minimal overlap in distribution with the ER marker CLIMP63 (Fig. 5c). Collectively, these results demonstrated that PolD1 is present at the Golgi complex.

To exclude the possibility that the nuclear localization of PolD1 is involved in its function of controlling Golgi-associated microtubule growth, we created a PolD1 truncation construct (247–955) that contains the DNA polymerase and exonuclease domains, but lacks the nuclear-localization signal[23]. This truncated protein robustly coimmunoprecipitated γ-tubulin from extracts of transfected cells, whereas the amino-terminal fragment of PolD1, 1–316, failed to coimmunoprecipitate γ-tubulin (Fig. 5d). Therefore, 247–955 retains the γTuRC-binding activity. Importantly, when expressed in cells, 247–955 appeared in the cytoplasm and exhibited no detectable nuclear localization (Fig. 5e), and the expression of 247–955 strongly inhibited microtubule regrowth at the Golgi complex (Fig. 5e). These results, together with those above (Fig. 5b, c), strongly suggest that PolD1 acts at the Golgi in the control of microtubule nucleation.

**PolD1 regulates Golgi organization and assembly**. In mammalian cells, the Golgi complex is a major microtubule-organizing center in addition to centrosomes[1, 2]. Golgi-associated microtubules are indispensable for the proper assembly and maintenance of the Golgi structure, and proper Golgi assembly depends on the coordinated actions of Golgi- and

centrosome-derived microtubules[5, 6, 27]. Here, we silenced PolD1 expression or overexpressed PolD1 and evaluated the effects on Golgi structure, and we also performed γ-tubulin staining to label centrosomes. Notably, PolD1 knockdown caused an expansion of the Golgi region (by ~52%) and reduced the compactness of the Golgi (by ~15%; Fig. 6a). Furthermore, the knockdown increased the density of microtubules in the Golgi region (by ~57%; Fig. 6a). Conversely, PolD1 overexpression considerably reduced the size of the Golgi complex (Fig. 6b). However, neither the knockdown nor the overexpression affected Golgi positioning relative to the centrosomes (Fig. 6a, b). Collectively, these findings indicate that the degree of Golgi expansion and compactness is regulated by cytoplasmic PolD1 levels. In RPE1 cells, the PolD1 level varies even among cells of the same cell cycle stage[28], and we found that a small fraction of interphase RPE1 cells (~5%) expressed PolD1 at significantly diminished levels; notably, in these cells, Golgi-associated microtubule regrowth was increased in correlation with the reduced levels of PolD1, and the Golgi complex was again enlarged (Supplementary Fig. 6).

Next, we evaluated the role of PolD1 in Golgi reassembly after nocodazole-induced disassembly. In agreement with published results[5], Golgi ministacks were gathered into clusters in the cytoplasm rapidly after nocodazole washout (G-stage, Fig. 6c), after which the clusters translocated toward the centrosome, where they were assembled into a Golgi ribbon (C-stage, Fig. 6c). At 1 h post washout, an intact Golgi ribbon was assembled around the centrosome (Fig. 6c). In the PolD1-depleted cells, Golgi clusters were formed in the cytoplasm as rapidly as in the control cells (Fig. 6c), which showed that the G-stage of Golgi reassembly was not affected by PolD1 depletion. However, the C-stage assembly was markedly impaired in the PolD1-depleted cells, and after 1 h of reassembly, the total amount of the Golgi particles that were unassociated with the Golgi and were dispersed in the cytoplasm was ~21% higher in the PolD1-depleted cells than in control cells (Fig. 6c). The excessive Golgi-derived microtubules interfered with the translocation of the Golgi ministacks toward the centrosome and the integration of the ministacks into the Golgi ribbon. Therefore, a balance of the actions produced by Golgi- and centrosome-derived microtubules is essential for proper Golgi reassembly. This supports the notion that proper Golgi assembly requires a concerted effort of these two microtubule populations[5, 27, 29].

**PolD1 functions in Golgi reorientation and cell migration**. Because the Golgi complex and its associated microtubules play a pivotal role in cell polarization and directional migration, we performed a wound-healing assay, and we first labeled Golgi complexes, as well as their associated centrosomes, to investigate the role of PolD1 in Golgi reorientation. After scratch-wounding, cells located at the scratch border were examined. At the beginning of wound healing, similar fractions of control and PolD1-depleted cells showed their Golgi oriented toward the leading edge (~25% of the cells; Fig. 7a). Over the next few hours,

**Fig. 6** Alteration of PolD1 expression affects Golgi organization. **a** RPE1 cells were transfected with control or PolD1 siRNAs. Anti-α-tubulin fluorescence intensities were measured at the Golgi area, and the background intensity was obtained from cytoplasmic areas lacking microtubules. Here, the average microtubule intensities per pixel at the Golgi are presented after subtraction of the background. The Golgi area was outlined according to TGN46 staining by using the freehand selection option in ZEN software. Data are shown as means ± s.d. of three experiments; $**p < 0.01$, $*p < 0.05$, two-tailed, unpaired student's *t*-test; control, $n = 60$ cells; PolD1 depletion, $n = 50$ cells per experiment. **b** Cells were transfected with GFP or GFP-PolD1 and then immunostained for GM130 and GCP6; $n = 50$ cells from three independent experiments. **c** RPE1 cells transfected with control or PolD1 siRNAs were treated with nocodazole. After nocodazole washout, Golgi reassembly was examined at the indicated time points. TGN46 immunofluorescence was used to measure the size of the Golgi particles that were not clustered in the Golgi region surrounding the centrosome. Quantification graphs show the average size of the individual particles and the total area of the particles. Data are presented as means ± s.e.m. of three experiments (30 cells analyzed per condition). Scale bars, 10 μm

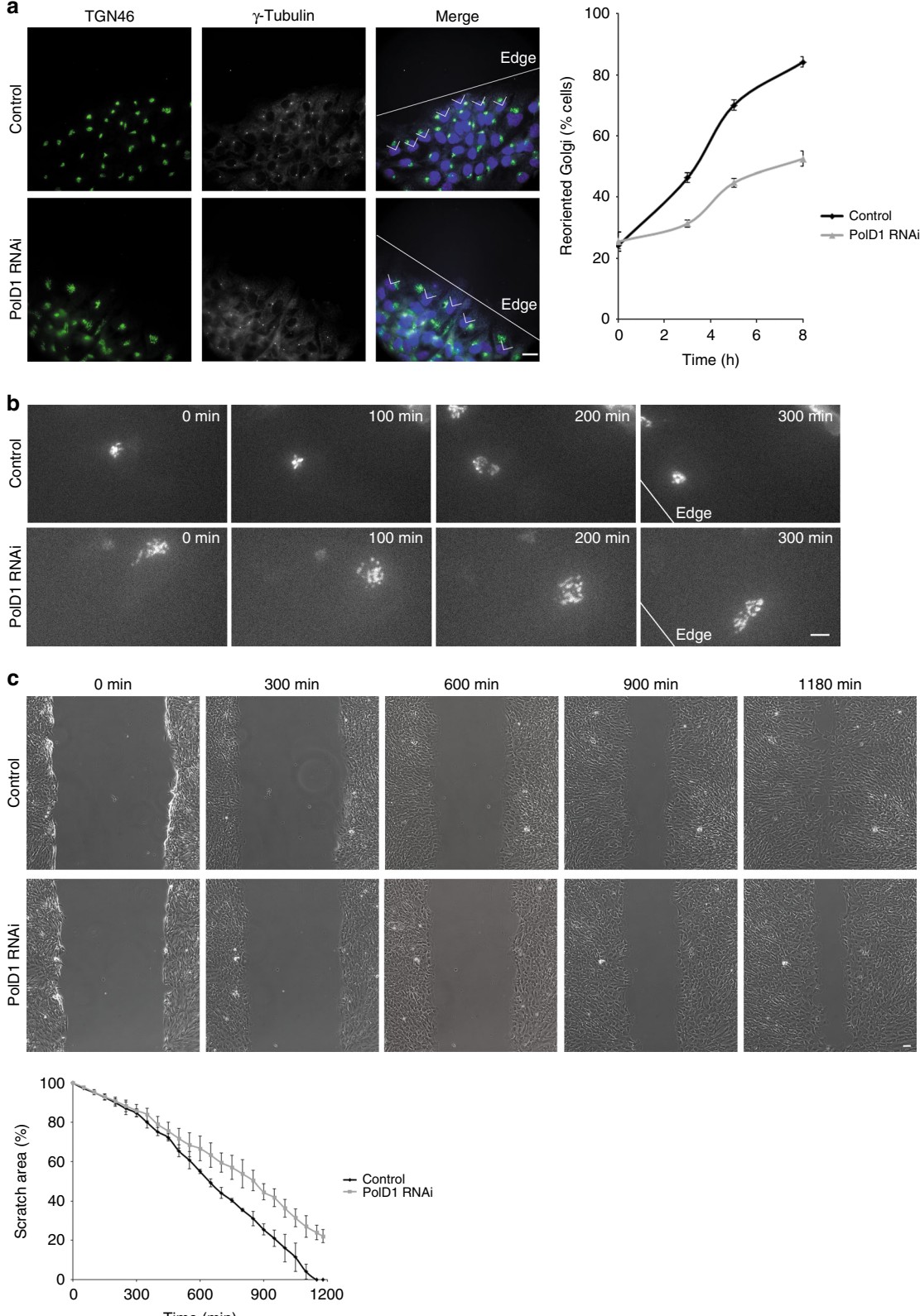

**Fig. 7** Cell polarization is impaired in PolD1-depleted cells. **a** In a wound-healing assay, RPE1 cells were immunostained to analyze Golgi reorientation. DNA was stained with Hoechst 33258. *White lines* mark the scratch edges. The angles indicate orientation within 90° facing the scratch edge. Numbers of cells containing the reoriented Golgi are plotted as means ± s.e.m., and the graph is representative of results obtained in three independent experiments (*n* = 200 cells for each condition). *Scale bar*, 20 μm. **b** Movie frames of RPE1 cell imaging were selected at the indicated time points after scratching. The cells were transfected with Venus-β-1,4-galactosyltransferase. Shown are representatives of 30 observed cells per condition. *Scale bar*, 10 μm. **c** A wound-healing assay was performed using RPE1 cells; selected phase-contrast movie frames are shown. The percentages of scratch area at each time point are shown as means ± s.e.m. The graph is representative of results obtained in three independent experiments. *Scale bar*, 20 μm

the PolD1-depleted cells showed drastically lower fractions with the reoriented Golgi than did the control cells; at 8 h post wounding, ~52% of the PolD1-depleted cells and ~84% of the control cells contained the reoriented Golgi (Fig. 7a).

To gain insights into the Golgi reorientation process, we performed time-lapse imaging of cells ectopically expressing the Golgi marker β-1,4-galactosyltransferase[30]. In control cells, the reorientation started with a slight Golgi fragmentation, and this was followed by the migration of the Golgi fragments to the nuclear side facing the leading edge and also to the site where the Golgi was eventually reassembled (Fig. 7b and Supplementary Movie 1). This agrees with the observations presented in a recent report[6]. Golgi fragmentation and reorganization are recognized to be required for Golgi reorientation[6]. The Golgi in PolD1-depleted cells was less compact than that in control cells, and it mostly maintained its structure during reorientation (Fig. 7b and Supplementary Movie 2). Consequently, the reorientation was considerably slower in the PolD1-depleted cells (Fig. 7a and Supplementary Movies 1 and 2). These data revealed that the ability of the Golgi to reorganize during reorientation is impaired in the absence of PolD1.

Lastly, we monitored cell migration in the wound-healing assay. Notably, PolD1 depletion reduced cell migration into the wound (Fig. 7c and Supplementary Movies 3, 4). By contrast, in an assay of cell random migration, the suppression of PolD1 expression did not discernibly affect either directional persistence or cell migration distance (Supplementary Fig. 7). Taken together, these results suggest that PolD1 is not required for the random migration of cells, but regulates cell polarization and directional migration.

## Discussion

As the principal microtubule nucleators in cells, γTuRCs are recognized to play a key role in the organization of microtubules[9–11]. Unexpectedly, however, little is known about the mechanisms that control the microtubule-nucleating activity of γTuRCs. Here, we have revealed a previously unappreciated aspect of γTuRC regulation and have also demonstrated a previously unknown function of PolD1, a conserved protein that is widely recognized for its role in DNA replication and repair[20–23]. We have shown that PolD1 physically associates with γTuRCs and thereby blocks γTuRC-induced microtubule nucleation, and that this PolD1 action is required for the control of microtubule growth at the Golgi complex. Consequently, PolD1 regulates several Golgi activities that involve Golgi-derived microtubules, including Golgi assembly and structural organization, Golgi reorientation during cell polarization, and cell directional migration.

Our results show that PolD1 acts directly on γTuRCs and potently inhibits the microtubule-nucleating activity of both CDK5RAP2-stimulated and CDK5RAP2-free γTuRCs. Given that distinct cytoplasmic γTuRC populations exist[31], our data suggest that PolD1 functions as a universal γTuRC inhibitor. PolD1 has been shown to exist in a monomeric form and in a heterodimeric form with PolD2, in addition to being present in the Pol δ heterotetramer[32]. Here, in our mass spectrometric analyses, PolD1 was identified with high confidence in the isolated γTuRCs, but no other Pol δ subunit was detected. Thus, the PolD1 that is not bound to any other Pol δ subunit most likely controls γTuRC activities. We also found that PolD2 does not inhibit γTuRC activity (Fig. 2c). The inhibition of γTuRCs could therefore represent a function unique to PolD1 among all Pol δ proteins.

In animal cells, a substantial amount of γ-tubulin is assembled into γTuRCs that exhibit localization-dependent microtubule nucleation activity[16]. Based on considering the data on PolD1 abundance in mammalian cells and the size of these cells[33–36], we estimated that the cellular PolD1 concentration is in the range of tens of nanomolar, which is close to the measured $IC_{50}$ of PolD1 toward γTuRCs. Because a majority of the cellular PolD1 exists in the cytoplasm (Fig. 5a), we suggest that PolD1 likely confers a highly effective cytoplasmic control of γTuRC activities, and further that this PolD1-mediated control of γTuRC activities is conserved at least in animal cells.

In this study, we have revealed the presence of PolD1 at the Golgi complex (Fig. 5b, c), and we propose that PolD1 acts on γTuRCs localized at the Golgi complex and perhaps at other noncentrosomal sites. Whereas the disruption of PolD1 expression did not affect centrosomal microtubule nucleation, it stimulated microtubule nucleation at the Golgi complex (Fig. 4), which is the most prominent noncentrosomal microtubule-organizing site in RPE1 cells. Therefore, our results reveal a mechanism that controls noncentrosomal γTuRC activities. This also raises the possibility that PolD1 dissociates from γTuRCs upon γTuRC recruitment to centrosomes.

By manipulating the level of PolD1, we set the experimental conditions required to alter the growth of Golgi-derived microtubules, and we found that the induced changes in Golgi-derived microtubules directly affected the Golgi structure. The suppression of PolD1 expression promoted Golgi-associated microtubule growth and resulted in an increase in the size and a decrease in the compactness of the Golgi complex. By contrast, PolD1 overexpression induced Golgi structural defects that were highly similar to those produced by RNAi-mediated depletion of AKAP450 or CLASPs, which were shown to remove Golgi-derived microtubules[2, 3, 6]. Collectively, these findings indicate that PolD1 participates in the organization of the Golgi structure by controlling Golgi-associated microtubule growth. Moreover, our results indicate a crucial role of Golgi-derived microtubules in determining the degree of Golgi ribbon extension and compactness, which is reminiscent of the action of Golgi-associated F-actin[37]. Therefore, the organization of an extended Golgi architecture requires both its associated cytoskeletons.

Intriguingly, PolD1 depletion caused defects in Golgi assembly and reorientation and cell directional migration. Currently, Golgi-derived microtubules are considered to act in a coordinated manner with centrosome-derived microtubules in the assembly and reorientation of the Golgi complex[5, 27, 29]. Given that the PolD1 depletion did not markedly affect centrosome-based microtubule growth, the increased growth of Golgi-derived microtubules would have changed the relative proportion of centrosome- and Golgi-derived microtubules. Thus, our finding suggests that the maintenance of a balance between these two microtubule populations is required for proper Golgi assembly and reorientation.

Microtubules are also nucleated at the Golgi membrane during mitosis, but at highly diminished levels as compared with those at the interphase Golgi complex[38]. PolD1 expression displays cell cycle-dependent changes and peaks at G2/M[28]. Upon mitotic entry and thus the breakdown of the nuclear envelope, the release of nuclear PolD1 into the cytoplasm and the elevated expression of PolD1 might contribute to the reduction of the microtubule-growing capacity at the Golgi. Overall, our findings demonstrate the existence of a PolD1-mediated mechanism by which γTuRC-induced microtubule growth at the Golgi complex, and perhaps at other noncentrosomal locations, is controlled. PolD1 is a ubiquitously expressed protein that performs housekeeping functions in the nucleus. Our study has revealed another critical function of PolD1, which the protein performs in the cytoplasm.

## Methods

**Plasmids and oligonucleotides**. The human PolD1 clone (GenBank Accession No.: NM_001256849) was obtained from the DNASU Plasmid Repository. A mutation (R119H) found in the clone was corrected using site-directed

mutagenesis. A bacterial expression construct of the PolD1 mutant D515V (exo-1) was a gift from Dr. Yoshihiro Matsumoto (The University of New Mexico, USA)[26]; the in-frame deletion mutant S605del in a bacterial expression plasmid was a gift from Dr. Larry Loeb (University of Washington, USA)[25]. PolD1 was subcloned using standard cloning techniques. The biotinylation (Bio)-2 × TEV-EGFP-C1 plasmid vector was provided by Dr. Anna Akhmanova (Utrecht University, Netherlands)[39]. Three siRNA oligonucleotides targeting *pold1* were synthesized: 5′-GUGCCAAGGUGCAGAGCUA-3′ (PolD1 RNAi-1), 5′-GAGAGAGCAU-GUUUGGGUA-3′ (PolD1 RNAi-2), and 5′-GGGACCAGGGAGAAUUAAU-3′ (PolD1 RNAi-3). Identical results were obtained with the three siRNAs, and those obtained with PolD1 RNAi-2 are presented here, except in the case of the rescue experiments, which were performed using PolD1 RNAi-3 (which targets the untranslated region). The siRNA duplexes targeting *gcp4* (5′-GCAAUCAA-GUGGCGCCUAA-3′) and *akap450* (5′-AACUUUGAAGUUAACUAUCAA-3′) were synthesized as previously described[3, 4, 17].

**Recombinant proteins and antibodies**. To prepare PolD1 and PolD2 proteins, HEK293T cells were cotransfected with biotin ligase BirA and Bio-2 × TEV-FLAG-PolD1, PolD1 mutants, or PolD2. The cells were extracted in a lysis buffer (50 mM HEPES, pH 7.2, 150 mM NaCl, 1 mM EGTA, 1 mM MgCl$_2$, 1 mM dithiothreitol, and Complete Protease Inhibitor Cocktail [Roche]) containing 0.5% IGEPAL CA-630 (Sigma-Aldrich) and 8 U ml$^{-1}$ Universal nuclease (Thermo Fisher Scientific). After clarification through centrifugation at 100,000 × *g* (30 min at 4 °C), the extracts were used in pulldowns performed with streptavidin-coupled paramagnetic beads (Promega). The pulldown samples were extensively washed in lysis buffer plus 1 M NaCl and 0.1% IGEPAL CA-630, and subsequently washed with lysis buffer plus 0.01% IGEPAL CA-630. After washing, PolD1 was released from the beads by means of cleavage with TEV protease and then desalted by passing through a Zeba Spin column (40-kDa cutoff; Thermo Fisher Scientific) pre-equilibrated in lysis buffer containing 0.01% IGEPAL CA-630. The purified proteins were verified through Coomassie Blue staining and immunoblotting.

Recombinant proteins containing His$_6$ or GST tags were expressed in *Escherichia coli* BL21 (DE3) and purified using Ni$^{2+}$-nitrilotriacetic acid resin (Qiagen) or glutathione-agarose beads (Sigma-Aldrich), respectively, dialyzed against phosphate-buffered saline supplemented with 10% glycerol, and then stored at −80 °C. Anti-PolD1 antisera were generated by immunizing rabbits with His$_6$-tagged PolD1(1–316), and the antibody was purified using the GST-tagged antigen immobilized on nitrocellulose membranes. A goat anti-PolD1 antibody was purchased from Santa Cruz Biotechnology (1:500 dilution, cat. no. sc-8797). The two antibodies yielded identical results in our experiments. The production and purification of the following antibodies (1:500 dilution) have been described previously: rabbit polyclonal anti-GCP2, anti-GCP3, anti-GCP4, anti-GCP5, anti-GCP6, anti-CDK5RAP2, and anti-AKAP450[4, 17, 18]. Antibodies recognizing CLASPs (1:300 dilution) were gifts from Drs. Anna Akhmanova (Utrecht University, The Netherlands), Niels Galjart (Erasmus Medical Center, The Netherlands), and Fedor Severin (Moscow State University, Russia)[40, 41]. These antibodies were purchased: anti-FLAG (1:1000 dilution, monoclonal M2, Sigma-Aldrich, cat. no. F1804), anti-GAPDH (1:1000 dilution, Thermo Fisher Scientific, cat. no. AM4300), anti-Histone H3 (1:1000 dilution, Abcam, cat. no. ab176842), anti-γ-tubulin (1:1000 dilution, GTU88, Sigma-Aldrich, cat. no. T5326), anti-α-tubulin (1:500 dilution, YL1/2, Santa Cruz Biotechnology, cat. no. sc-53029), anti-GM130 (1:500 dilution, monoclonal, BD Biosciences, cat. no. 610822), anti-TGN46 (1:500 dilution, sheep polyclonal, Serotec, cat. no. AHP500), anti-PolD2 (1:500 dilution, Santa Cruz Biotechnology, cat. no. sc-8800), and DyLight or Alexa Fluor secondary antibodies (1:500 dilution, Thermo Fisher Scientific).

**γTuRC purification**. HEK293T cells were used for γTuRC isolation because of their robust growth and ease of transfection. γTuRCs were isolated through tandem affinity purification from a 1:1 mixture of FLAG-2 × TEV-SBP-CDK5RAP2 (1–100)-transfected and untransfected HEK293T cells. The cell extracts were prepared in lysis buffer containing 0.5% IGEPAL CA-630 and 0.2 mM GTP and then clarified. Anti-FLAG immunoprecipitation was performed by incubating the extracts with anti-FLAG agarose (Biotool) at 4 °C for 2 h. The beads were washed with lysis buffer plus 0.5% IGEPAL CA-630 and subsequently with the buffer plus 0.01% IGEPAL CA-630, and then SBP-CDK5RAP2(1–100) and its bound proteins were released by means of cleavage with 25 ng ml$^{-1}$ TEV protease at 4 °C for 2 h. The obtained supernatants were used in pulldowns performed with streptavidin-coupled beads, and the bound proteins were eluted by incubating the beads with 10 mM biotin (Santa Cruz Biotechnology) for 30 min. The eluate was passed through a Zeba Spin column (40-kDa cutoff) pre-equilibrated in lysis buffer containing 0.01% IGEPAL CA-630.

**Microtubule nucleation in vitro**. The in vitro nucleation assays were performed at 37 °C for 5 min as described[17, 19]. Before use in the assays, purified γTuRCs were incubated at 4 °C for 30 min with CDK5RAP2(51–200) (28 nM), PolD1, and PolD2 in different combinations. After nucleation, the microtubules were fixed in 1% glutaraldehyde and diluted. An aliquot of the diluted samples was centrifuged (173,000 × *g*, 8 min) in tubes (TLS55; Beckman Coulter) equipped with a Teflon platform to sediment microtubules through a cushion of 15% glycerol/BRB80

(80 mM PIPES-KOH, pH 6.8, 1 mM EGTA, and 1 mM MgCl$_2$) onto a coverglass. For each sample, microtubules were counted in 10 randomly selected fields under a Carl Zeiss microscope (Axio Observer Z1).

**Immunoprecipitation and in vitro binding assays**. For immunoprecipitations, cell extracts were prepared at 4 °C in a lysis buffer containing 0.5% IGEPAL CA-630 and clarified, and then Protein A-agarose (Thermo Fisher Scientific) coupled with primary antibodies or anti-FLAG M2-coupled beads (Sigma-Aldrich) were added into the extracts and incubated at 4 °C for 2 h. Subsequently, the beads were extensively washed in lysis buffer containing 0.01% IGEPAL CA-630 and the bound proteins were analyzed by immunoblotting the obtained samples. To test protein binding in vitro, FLAG-PolD1 was incubated with purified γTuRCs in lysis buffer containing 0.01% IGEPAL CA-630 and 1 mg ml$^{-1}$ bovine serum albumin (Sigma-Aldrich) at 4 °C for 2 h. After incubation, anti-FLAG immunoprecipitation was performed.

**Mass spectrometry analysis and data processing**. γTuRCs were isolated by anti-FLAG immunoprecipitation as described above and without streptavidin pulldown. After anti-FLAG immunoprecipitation, bound proteins were eluted by incubating the beads with FLAG peptide (200 ng μl$^{-1}$)[17, 19] and the eluted proteins were resolved using SDS-PAGE and stained with SYPRO Ruby (Thermo Fisher Scientific). Protein bands excised from the SDS-PAGE gels were digested with trypsin, and the extracted peptides were analyzed using mass spectrometry (LTQ Velos linear ion-trap LC–MS system, Thermo Fisher Scientific). The acquired tandem mass spectra were subject to gene database searches by using the MASCOT search engine (Matrix Science).

**Nuclear extraction**. Nuclear and cytoplasmic proteins were fractionated according to the method of Qu et al.[42]. Briefly, cells were homogenized in nucleus buffer (10 mM Tris-HCl, pH 7.4, 10 mM KCl, 2 mM MgCl$_2$, 1 mM dithiothreitol, 30% sucrose, and Complete Protease Inhibitor Cocktail), and after centrifugation at 1000 × *g* for 10 min at 4 °C, the supernatant was collected as the cytoplasmic fraction. The pellet was washed with nucleus buffer and then collected as the nuclear fraction.

**Golgi isolation**. Golgi membranes were isolated at 4 °C using the equilibrium gradient method[43, 44]. RPE1 cells were homogenized in homogenization buffer (10 mM Tris-HCl, pH 7.4, 0.25 M sucrose, 5 mM EDTA, and Complete Protease Inhibitor Cocktail) and the extracts were centrifuged at 1000 × *g* for 10 min. After centrifugation, the PNS was collected. To prepare the sucrose gradient, five sucrose layers (450 μl of 1.6 M, 450 μl of 1.4 M, 450 μl of 1.2 M, 350 μl of 0.8 M, and 250 μl of 0.5 M sucrose in 10 mM Tris-HCl and 5 mM EDTA buffer, pH 7.4) were sequentially overlaid from the bottom to the top of a centrifuge tube (TLS55 rotor, Beckman). The PNS (250 μl) was carefully overlaid atop this gradient and centrifuged at 90,000 × *g* for 3.5 h, after which the gradient was collected into 12 fractions from the top to the bottom. Each fraction was diluted by threefold in 10 mM Tris-HCl (pH 7.4) and centrifuged at 180,000 × *g* for 1 h, and the pellets (isolated membranes) were then analyzed by immunoblotting.

**Cell culture**. HEK293T and RPE1 cells (purchased from American Type Culture Collection as cell lines CRL-11268 and CRL-4000, respectively) were cultured at 37 °C and 5% CO$_2$ in a humidified atmosphere with the following media: HEK293T cells, Dulbecco's modified Eagle's medium (DMEM, Gibco) containing 10% fetal bovine serum (Gibco) and 1% penicillin/streptomycin; RPE1 cells, DMEM/Ham's F12 (1:1) containing 10% fetal bovine serum, 1% penicillin/streptomycin, and 10 μg ml$^{-1}$ hygromycin B (Sigma-Aldrich). These cell lines were authenticated using short tandem repeat profiling analysis. Cell cultures were free of mycoplasma contamination. Plasmid and siRNA transfections were performed using Lipofectamine 2000 (Thermo Fisher Scientific), FuGENE (Promega), or polyethylenimine (Polysciences)[19].

**Immunofluorescence microscopy**. Immunostaining was performed on RPE1 cells that were grown on poly-D-lysine-coated coverslips. Cells were fixed with 4% paraformaldehyde in PHEM buffer (60 mM PIPES-KOH, 25 mM HEPES, pH 6.9, 10 mM EGTA, and 2 mM MgCl$_2$) containing 0.5% Triton X-100 for 15 min at room temperature or with methanol for 10 min at −20 °C. After sequential staining with primary antibodies and Alexa Fluor-conjugated secondary antibodies, fluorescence images were acquired using an epifluorescence microscope (Axio Observer Z1, Carl Zeiss) equipped with an ORCA-Flash 4.0 camera (Hamamatsu Photonics). Images were acquired and processed using ZEN microscope software (Version 2011, Carl Zeiss). To visualize PolD1 at the Golgi complex, cells were extracted in saponin buffer (100 mM PIPES-KOH, pH 6.9, 2 M glycerol, 5 mM MgCl$_2$, 2 mM EGTA, and 1% saponin) for 30 min and then fixed in methanol at −20 °C for 5 min. The cells were stained with a goat anti-PolD1 antibody and then sequentially stained with two secondary antibodies: Alexa Fluor-conjugated donkey anti-goat and DyLight Fluor-conjugated rabbit anti-donkey antibodies (Thermo Fisher Scientific).

 

To examine microtubule regrowth, cellular microtubules were first depolymerized by placing cells on ice water for 1 h. Microtubule regrowth was initiated at 37 °C and terminated at various time points by fixing the cells. Golgi-associated microtubules were quantified and normalized by the Golgi mass, which was determined based on the total fluorescence intensity of GM130 within the Golgi area. Image analysis was performed using ImageJ software.

Golgi compactness was determined by analyzing the TGN46 staining signal by using a reported method[27]. Briefly, the Golgi contour was drawn by using the freehand selection option of ZEN software, and the TGN46 signal within this area was detected using an ImageJ plugin for particle analysis. Golgi compactness was calculated as the percentage of the area occupied by the TGN46 signal (Golgi stacks) inside the contoured area.

In Golgi reassembly assays, cells were incubated with 2.5 μg ml$^{-1}$ nocodazole at 37 °C for 3 h and then incubated on ice for 1 h. After washing with ice-cold medium, Golgi reassembly was initiated by transferring the cells to prewarmed medium and was terminated at various time points by fixing the cells. The size of individual Golgi particles dispersed in the cytoplasm was measured using ImageJ software. In wound-healing assays, cells seeded on coverslips were allowed to form a confluent monolayer and then starved for 24 h. A scratch was made on the monolayer cells with a pipette tip. After 20-min recovery, the cells were placed in medium containing 10% fetal bovine serum at 37 °C to induce migration. Golgi reorientation was analyzed in cells at the wound edge.

**Time-lapse microscopy**. In Golgi reorientation and directional migration experiments, cells transfected with Venus-β-1,4-galactosyltransferase were imaged on an inverted microscope (Nikon Ti-E-PFS microscope) equipped with an sCMOS camera (Zyla ultra-low noise sCMOS camera, Andor). To monitor Golgi reorientation, images were captured at 5-min intervals for 8 h. Cell directional migration was analyzed from images captured at 10-min intervals for 20 h. To analyze random migration, cells were seeded at a low density and images were captured at 5-min intervals for 3 h. Image processing and analysis were performed using MetaMorph and ImageJ.

**Image processing**. All fluorescence images shown are wide-field microscopy images. Brightness and contrast were adjusted individually for each fluorescence channel. No gamma settings were changed in any of the images. All images were cropped and/or enlarged to show representative examples in sufficient detail. Uncropped blots are shown in Supplementary Fig. 8.

**Statistics and reproducibility**. All results presented in graphs are means ± s.e.m. or s.d., as indicated in figure legends. The exact $n$ values are indicated in the corresponding figure legends. At least three independent experiments were performed for each condition. No statistical method was used to predetermine sample size. No samples were excluded from the analyses, and samples were not randomized. The investigators were not blinded to allocation during experiments and outcome assessment. To the best of our knowledge, the statistical tests are justified as appropriate. Immunoblotting and immunofluorescence data presented are representative of at least three independent experiments that yielded similar results.

**Data availability**. All source data for graphs and statistical analysis are presented in Supplementary Tables 2–16. All relevant data that support the conclusions of the study are available from the authors on request.

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

## Acknowledgements

We thank Dr. Yanzhuang Wang (University of Michigan, Ann Arbor, MI) for advice on Golgi membrane isolation and Dr. Bik-Kwoon Tye (The Hong Kong University of Science and Technology) for critical reading of the manuscript. This work was supported by grants from the Research Grants Council (General Research Fund and Theme-based Research Scheme) of Hong Kong, the National Key Basic Research Program (2013CB530900) and the Shenzhen Peacock Plan of China, the University Grants Committee (Area of Excellence Scheme) of Hong Kong, the Innovation and Technology Commission (ITCPD/17-9) of Hong Kong, and the TUYF Charitable Trust. R.Z.Q. is affiliated with Guangdong Provincial Key Laboratory of Brain Science, Disease and Drug Development, HKUST Shenzhen Research Institute, Shenzhen, China.

## Author contributions

R.Z.Q., Y.S. and P.L. designed the experiments. Y.S., P.L., T.J. and F.K.C.A. performed the experiments. Y.H. and T.J. performed γTuRC isolation for mass spectrometry. R.Z.Q. and Y.S. wrote the manuscript, with input from all authors.

## Additional information

**Competing interests:** The authors declare no competing financial interests.

