## [Peer Review File · Nature Communications]

Reviewers' Comments:

Reviewer #1 (Remarks to the Author)

This manuscript presents an exciting finding of a novel factor that regulates MT nucleation, with a stronger effect on Golgi-derived MTs. The data are substantial including in vitro reconstitution, cellular and functional assays.

Major concerns:

1. The authors show that cells depleted of PoD1 possess enlarged Golgi structures and increased GDMT levels, while cells overexpressing PoD1 possess few GDMTs and a smaller Golgi. However, the authors do not investigate whether alterations in PoD1 levels lead to altered Golgi structures (e.g. due to disturbance of membrane trafficking) and thus altered GDMT levels, or whether PoD1 levels affect GDMT levels, which in turn then affect Golgi size.

2. GDMT number per cell and per Golgi area/mass should be quantified in all experiments.

3. How does overexpressing of PoD1 compare to its endogenous localization pattern? Does endogenous PoD1 localize at the Golgi, and what are normal ratios of nuclear:cytosolic protein? Figure 2c shows GFP-PoD1 accumulations in the cytosol (not at the Golgi), where/what are these accumulations? In later figures these accumulations do appear present when immunostaining for endogenous PoD1, can the authors clarify whether overexpression represents an amplification of the endogenous localization or if the localization is altered when overexpressed? If there is no PoD1 at the Golgi (like the immunofluorescence images in Figure 3 suggest), are there known downstream effectors of PoD1 outside of the nucleus that may help to explain the observations outlined here?

4. The authors should expand on the functional relevance and mechanism of the PoD1-mediated reduction in GDMT levels. Where is PoD1 localized during interphase and leading up to mitosis, when GDMT levels are higher than during mitosis? The authors mention that PoD1 is enriched in the nucleus during interphase, but do not mention whether it is at all present in the cytosol or at the Golgi during this time, though the PoD1-depletion experiments would suggest there is at least some PoD1 present in the cytosol and/or at the Golgi.

5. Do the depletion experiments mimic/exaggerate the interphase state, and do the overexpression experiments mimic/exaggerate the mitotic state? The authors show that PoD1 interacts with γ -TuRCs, but that overexpression does not alter centrosomal γ -TuRC levels. Can the authors expand on possible mechanisms whereby PoD1 may regulate GDMTs? Are γ -TuRCs sequestered in the cytosol? Are other factors associated with GDMT nucleation and stabilization (CDK5RAP2, AKAP450, CLASPs etc) altered at the Golgi? Moreover, the authors do not discuss the templated nucleation model proposed by Wieczorek et al. in 2015. Figure 1 shows that the γ -TuRC is not disrupted upon PoD1 depletion, so it is possible that subsequent steps (tubulin recruitment, elongation, stabilization) could be affected by PoD1, not the initial nucleation step.

Minor concerns:

Pg. 3 line 10-11. Which other proteins were detected in the mass spectrometry? If the authors do not wish to release the full list, the peptide count for PoD1, cut-offs used to remove false positive results etc. should be included. If this list was previously published, authors should reference this.

Pg. 3 line 19 – pg. 4 line 2. These pull down experiments should also be performed with

overexpressed PoID1 as many of the later experiments use this as a method to show PoID1 regulates GDMT levels. Does PoID1 overexpression disrupt the γ -TuRC?

Pg. 4 line 2-3. Were other Pol δ subunits pulled down with transiently expressed PoID1? Control experiment should be included to show that the construct does not affect its known interactions.

Pg. 4 line 7-8 Figure 1. Quantification of reduced interactions? As a control, can the authors show that the γ -TuRC is actually disrupted by looking at other components, or is only GCP4 missing and/or are γ -TuSCs still intact?

Pg. 5 line 13-15. How do the IC50s relate to native conditions? What is the stoichiometry of γ -TuRC to PoID1?

Pg. 6 line 8-9. Is GDMT nucleation fully inhibited, or is it slower? The centrosomal asters in GFP-PoID1 expressing cells look smaller and the MTs look shorter. Are GDMTs still absent and centrosomal MTs shorter or slower-growing over time? A time series (or a quantification thereof) would answer this question.

Pg. 7 line 19-20. 'low levels', compared to what? How much lower than other cells? Are other factors associated with GDMT nucleation/stabilization also altered in these cells? How do alterations in expression level affect localization? Is PoID1 present in the cytosol or at the Golgi in these low-expressing cells?

Pg. 8 line 17. Write out '2'.

Pg. 9 line 15-16. Is Golgi orientation completed at all in PoID1 depleted cells eventually? Time series should be extended to determine this.

Pg. 10 line 13-14. Is PoID1 found at the Golgi?

Pg. 10 line 17. This is cell-type specific and should be mentioned as such.

Pg. 11 line 2. Write out '2'.

All figures showing fluorescence imaging: How are images shown here? Are they single frames, Z-stack projections, etc.? How were they processed; were gamma- or other non-linear adjustments made anywhere?

Figure 1.

1d. Were equal amounts of protein loaded after the IP? The FLAG-PoID1 levels seem variable in the image on the right (IP FLAG), especially the S605del band, which looks smaller than the others.

1c-d should have a control lane where an anti-FLAG antibody was used to show that when FLAG is expressed (in control (1c) or vector (1d)) but is not linked to PoID1, the interaction with γ -TuRC components is absent. The authors should specify that for the FLAG-PoID1 WB an antibody against PoID1 was used, not against FLAG.

Figure 2.

2a. see comment above on 1c-d, FLAG control immunoblotting should be shown.

2b. The y-axis label should read γ -TuRCs with microtubules (%) or something to that effect. Were γ -TuRCs labelled in any way to determine this percentage? A (supplementary) image should be included of a control experiment without isolated γ -TuRCs but with CDK5RAP2 to show that under the used conditions no MT nucleation can occur without γ -TuRCs.

Figure 4. 4a. What does 'Golgi compactness was quantified as the proportion of TGN46-occupied

space in the Golgi area' mean? The Golgi area was outlined based on the TGN46 staining, yet this was also used to measure the compactness within that area?

4a. Why is γ -tubulin immunostaining shown? This panel is not referenced anywhere in the paper.

4b. Quantification of Golgi positioning should be added.

4c. Statistics should be added to the quantification graphs.

Are all scale bars 10 μ m? In 4b the scale bar looks to be shorter than that based on the nuclear sizes compared between the three data panels.

Figure 5. 5a-c. Statistics should be added.

5a. Why is the γ -tubulin panel included? It is not referenced anywhere in the manuscript.

Supplementary Figure 2. The FLAG-PoD1 image shows some signal in the three cells on the right.

Are these cells transfected and weakly expressing FLAG-PoD1? If so, authors should also include a non-transfected cell to show no effect on γ -tubulin levels at the centrosome.

Supplementary Figure 3 c. Quantification of GDMT levels should be added.

Methods.

Mass spectrometry section is missing.

Cell culture and immunofluorescence microscopy. 6th line of the section, a '/' is missing between penicillin and streptomycin.

Reviewer #2 (Remarks to the Author)

General comments:

In this manuscript, Shen et al. extend their previous work on isolating the gamma-tubulin ring complex (γ TuRC) and found PoD1, the catalytic subunit of the DNA polymerase δ , co-purified with γ TuRCs. Through co-immunoprecipitation and western blotting analysis, they confirmed that PoD1 is in a complex with intact γ TuRCs and this association requires neither its DNA polymerase nor exonuclease activities. The authors then demonstrated that, in the presence of recombinant PoD1 protein, Cdk5Rap2-stimulated microtubule nucleation by γ TuRCs was attenuated in vitro, which suggests that PoD1 is a γ TuRC inhibitor. PoD1 overexpression impaired microtubule nucleation at the Golgi and rendered the Golgi more compact. Conversely, PoD1 knockdown increased microtubule nucleation at the Golgi and caused the Golgi to expand. Consequently, PoD1 RNAi resulted in defects in Golgi reassembly after nocodazole washout and in Golgi reorientation in a scratch wound assay. Based on these results the authors concluded that PoD1 negatively regulates Golgi-derived microtubule nucleation by inhibiting γ TuRCs.

If correct, the findings of a Golgi-specific γ TuRC inhibitor would be novel. However, the authors need to demonstrate that the effects are specific and direct. The major concern is that, the conclusions rely on overexpression and knockdown of PoD1, which is a catalytic subunit of the DNA polymerase δ . Given the essential functions of DNA polymerases, it is to be expected that these manipulations will induce adverse effects to the cells. The authors need to experimentally

dissect the different functions of PolD1 and use rigorous controls to rule out indirect effects.

A second and more profound issue is the lack of mechanistic insights into how PolD1 might regulate γ TuRC at the Golgi. Conceptually, it is difficult to reconcile the nuclear localization and the well-established function of PolD1 in DNA replication/repair with a Golgi-specific activity. The authors co-immunoprecipitated PolD1 with γ TuRCs from detergent lysates, and it is not clear whether PolD1 was released from the nucleus during lysis. At a minimum, the authors need to localize endogenous PolD1 to Golgi membranes by immunofluorescence or EM, as well as subcellular fractionation (without detergent). Without this evidence, it is likely that co-purification and co-immunoprecipitation of PolD1 with γ TuRCs is induced during detergent lysis when nuclear PolD1 is released into the cytosol.

On balance, the authors' findings are interesting but too preliminary to conclude that PolD1 regulates microtubule nucleation at the Golgi. In its present form, the manuscript is rather descriptive and correlative. There are significant holes in the story from the perspectives of both biochemistry and cell biology that need to be addressed before the manuscript is suitable for publication.

Specific comments

Major:

1. Figure 2b, in vitro microtubule nucleation assay:

The authors recognized that co-IP of PolD1 with γ TuRCs does not indicate their direct interaction and thus sought to reconstitute its direct inhibition on γ TuRCs in vitro using purified components. Unfortunately, this set of experiments lack essential controls and are unconvincing. As the authors noted, the reaction actually contains a Cdk5Rap2 domain that binds and stimulates γ TuRCs. PolD1 reduced this stimulation, which could be due to a direct inhibition on γ TuRCs as the authors proposed, or caused by the added PolD1 that sequesters/competes the Cdk5Rap2 domain and/or interferes its activity to stimulate γ TuRCs. To show a direct inhibition of γ TuRCs, the authors have to only use purified γ TuRCs, tubulins and PolD1. Furthermore, they need to dissect whether PolD1 interacts with the Cdk5Rap2 domain or interferes its binding to γ TuRCs. The reaction should also include specificity controls such as a same epitope-tagged γ TuRC binding-deficient mutant of PolD1 or a catalytic subunit of other DNA polymerases. In addition, since the polymerase-deficient and exonuclease-deficient PolD1 mutants interact with γ TuRCs, do they also suppress nucleation?

2. Figure 2c, PolD1 overexpression:

As mentioned above, the distribution and localization of endogenous PolD1 needs to be determined. How much PolD1 is localized to the nucleus, the cytoplasm and the Golgi (if any) respectively? The authors can access the relative ratio by immunofluorescence and subcellular (nuclear/cytosol/membrane) fractionations. Also, how does this ratio change when PolD1 is overexpressed or down-regulated? This information is needed to interpret their data. In Figure 2c, for example, overexpression of GFP-PolD1, both wildtype and two mutants, causes defects in microtubule nucleation at the Golgi complex. However, the expression levels of control GFP vs GFP-PolD1 are not presented. In addition, control GFP localizes predominantly to the nucleus, which is not a proper control for overexpressed PolD1 that has much higher cytosolic localization. Again the question is one of specificity.

3. Figures 3-5, PolD1 knockdown:

Most phenotypic and functional readouts heavily rely on PolD1 knockdown. The authors need to dissect whether the phenotypes were due to a direct PolD1 function at the Golgi or due to indirect secondary defects caused by inhibition of DNA replication. One possibility is that PolD1-RNAi cells may be enriched at a certain cell cycle stage when microtubule nucleation activity is coincidentally up-regulated. To address this concern, in addition to showing a part of PolD1 is localized to the Golgi, the authors should specifically inhibit cytoplasmic/Golgi-localized pool of PolD1 and

investigate its outcomes. For instance, microinjections of PolD1 protein (since it works in vitro) and/or of an inhibitory antibody against PolD1 (if available) into the cytoplasm, or using a Golgi-targeting construct, etc. Alternatively and perhaps easier, and AID0PolD1 would at least allow measuring acute effects of PolD1 loss. In addition, can expression of RNAi-resistant forms of PolD1 wildtype and two mutants rescue the phenotypes?

4. Supplementary Figures 4:

What percentage of the cells have low levels of PolD1? If this occurs at a high frequency, one cannot tell if the phenotypes are due to loss-of-function by RNAi or inherent heterogeneity within a cell population. Based on the reference cited by the authors (Ref. 27), the levels of PolD1 almost double in G2 phase compared to G1. Can the authors synchronize cells and compare the phenotypes between the different cell cycle stages? Are the results consistent with observed phenotypes?

Minor:

1. Figure 1a: Do other subunits (PolD2, PolD3, PolD4) of the DNA polymerase δ also co-IP with γ TuRCs? In other words, is the interaction specific to PolD1?

2. The following statements (underlined) are an over-interpretation of the data, and have not been convincingly shown in the current manuscript and should be modified.

The second last paragraph line 10:

We have shown that PolD1 physically associates with γ TuRCs in the cytoplasm

-> The authors used whole cell lysate for the co-IP, which includes nucleoplasm where presumably the majority of PolD1 is present. They did not present any data demonstrating PolD1 is recruited from the cytoplasm to the Golgi membranes either.

The second last paragraph line 14:

This action of PolD1 occurs at the Golgi complex

-> Supporting data are not provided.

Reviewer #3 (Remarks to the Author)

Interesting and well-written manuscript describes a novel and unexpected role of the catalytic subunit of main replicative DNA polymerase delta in microtubule dynamics. POLD1 directly binds to gammaTuRCs and inhibits microtubule nucleation. Known enzymatic activities of pol delta, exonuclease and polymerase, are apparently not required for the association. Depletion of POLD1 specifically enhances these processes at the Golgi, while overexpression inhibits them, altering its organization and cell behavior. The evidence presented by the authors is convincing, though some of the results are supported only by images with no quantitative estimates of the level of effects (Figs. 2 and 3).

Several things need to be clarified.

Pol delta is not the most abundant protein in the cell, while components of microtubules are. It will be good to discuss the stoichiometry of the inhibitory processes.

There is some confusion with RNAi inhibition experiment (sFig 3 and Fig. 3). POLD1 is vitally required for cell growth. Images of cells show no pol delta detectable by immunostaining in the nuclei. The authors should describe what else happens with these cell lines in addition to changes in Golgi.

It should be better explained what is the mechanism of specificity of pol delta to gammaTuRCs in Golgi. Ideas how free catalytic subunit of the tight pol delta complex appears in cells should be presented.

The most harmful weakness is a very limited discussion of biological meaning and significance of the discovered effect.

Minor comments:

It is not clearly described, what was the source of isolated gammaTuRCs when pol delta was first detected.

Brief description of human cell lines and rationale for choosing the particular ones (two cancer lines, one immortalized normal eye epithelium) would be helpful.

Graphs in Fig. 4 occupy random positions. They should be labeled, so the relation to images will be clear.

Point-by-point responses to reviewers' comments:

Reviewer #1:

This manuscript presents an exciting finding of a novel factor that regulates MT nucleation, with a stronger effect on Golgi-derived MTs. The data are substantial including in vitro reconstitution, cellular and functional assays.

Major concerns:

1. The authors show that cells depleted of PolD1 possess enlarged Golgi structures and increased GDMT levels, while cells overexpressing PolD1 possess few GDMTs and a smaller Golgi. However, the authors do not investigate whether alterations in PolD1 levels lead to altered Golgi structures (e.g. due to disturbance of membrane trafficking) and thus altered GDMT levels, or whether PolD1 levels affect GDMT levels, which in turn then affect Golgi size.

Response:

As suggested by the reviewer, we performed experiments to examine protein trafficking; we used time-lapse imaging and monitored the trafficking of Venus-Rab6A-labeled vesicles from the Golgi to the plasma membrane in hTERT-RPE1 cells. RNAi-mediated suppression of PolD1 expression did not markedly affect the vesicle trafficking. Therefore, the Golgi expansion was not an effect of vesicle trafficking. In the revised manuscript, we have presented results demonstrating that PolD1 controls γ TuRC-mediated microtubule nucleation at the Golgi and thus the amount of Golgi-associated microtubules. Therefore, we propose that PolD1 regulates the size and compactness of the Golgi by altering the levels of Golgi-derived microtubules (Page 15, paragraph 3).

2. GDMT number per cell and per Golgi area/mass should be quantified in all experiments.

Response:

We have now quantified the Golgi-derived microtubules in all microtubule regrowth experiments and added the normalized data into the figures (Figures 3c, 4, 5e, and Supplementary Figure 3c).

3. How does overexpressing of PolD1 compare to its endogenous localization pattern? Does endogenous PolD1 localize at the Golgi, and what are normal ratios of nuclear:cytosolic protein? Figure 2c shows GFP-PolD1 accumulations in the cytosol (not at the Golgi), where/what are these accumulations? In later figures these accumulations do appear present when immunostaining for endogenous PolD1, can the authors clarify whether overexpression represents an amplification of the endogenous localization or if the localization is altered when overexpressed? If there is no PolD1 at the Golgi (like the immunofluorescence images in Figure 3 suggest), are there known downstream effectors of PolD1 outside of the nucleus that may help

to explain the observations outlined here?

Response:

To address these questions, we performed the following new experiments. (1) Subcellular fractionation: nuclear and cytoplasmic fractions contained ~37% and ~63% of PolD1, respectively. (2) Isolation of Golgi membranes: both PolD1 and γ -tubulin were detected in isolated Golgi membranes through immunoblotting. (3) Immunostaining of PolD1: PolD1 was visualized at the Golgi complex in addition to being detected in the nucleus. Therefore, our results show that a substantial amount of PolD1 exists in the cytoplasm, and further that PolD1 is present at the Golgi complex. These new data are included in the revised manuscript (Figure 5a–c).

We analyzed the nuclear:cytosolic ratio of overexpressed PolD1 by performing subcellular fractionation and immunoblotting, and found that the ratio was close to 1:1 for GFP-PolD1. Therefore, the overexpressed PolD1 exhibited a similar nucleocytoplasmic distribution as the endogenous protein. As a control, we analyzed GFP alone, and found that most of this protein was present in the cytoplasmic fraction after the subcellular fractionation. These data have been included in the manuscript (Supplementary Figure 5). In the staining experiment shown in the original Figure 2c (Figure 3a of the revised manuscript), GFP-PolD1 was found to be dispersed in the cytoplasm, in addition to being localized in the nucleus and at the Golgi. In the cells transfected with the GFP-only construct, most of the cytoplasmic GFP was extracted during cell fixation with 4% paraformaldehyde plus 0.5% Triton X-100, and thus little cytoplasmic localization of GFP was observed.

In summary, we have now provided the evidence to show that PolD1 localizes at the Golgi and that ectopically expressed PolD1 exhibits a similar nucleocytoplasmic localization pattern as the endogenous protein. We propose that PolD1 acts at the Golgi to inhibit γ TuRC-induced microtubule growth.

4. The authors should expand on the functional relevance and mechanism of the PolD1-mediated reduction in GDMT levels. Where is PolD1 localized during interphase and leading up to mitosis, when GDMT levels are higher than during mitosis? The authors mention that PolD1 is enriched in the nucleus during interphase, but do not mention whether it is at all present in the cytosol or at the Golgi during this time, though the PolD1-depletion experiments would suggest there is at least some PolD1 present in the cytosol and/or at the Golgi.

Response:

As stated above, we have further revealed the mechanism of PolD1 action by demonstrating the nucleocytoplasmic distribution of PolD1 and the Golgi localization of PolD1. Moreover, we created a truncated PolD1 construct that lacks the nuclear localization signal and thus does not localize to the nucleus, and we found that this PolD1 construct binds to γ TuRCs and retains the inhibitory activity toward Golgi-associated microtubule regrowth. Therefore, cytoplasmic PolD1 acts in the control of Golgi-associated microtubule growth, and this PolD1 action is independent of PolD1 nuclear localization. These new data have been included in the revised manuscript

(Figure 5). Together, our data strongly support the conclusion that PolD1 acts at the Golgi to inhibit γ TuRC-mediated microtubule nucleation.

5. Do the depletion experiments mimic/exaggerate the interphase state, and do the overexpression experiments mimic/exaggerate the mitotic state? The authors show that PolD1 interacts with γ -TuRCs, but that overexpression does not alter centrosomal γ -TuRC levels. Can the authors expand on possible mechanisms whereby PolD1 may regulate GDMTs? Are γ -TuRCs sequestered in the cytosol? Are other factors associated with GDMT nucleation and stabilization (CDK5RAP2, AKAP450, CLASPs etc) altered at the Golgi? Moreover, the authors do not discuss the templated nucleation model proposed by Wieczorek et al. in 2015. Figure 1 shows that the γ -TuRC is not disrupted upon PolD1 depletion, so it is possible that subsequent steps (tubulin recruitment, elongation, stabilization) could be affected by PolD1, not the initial nucleation step.

Response:

We have now shown that nuclear and cytoplasmic fractions contain ~37% and ~63% of PolD1, respectively (Figure 5a), which indicates that a substantial proportion of PolD1 is present in the cytoplasm even in interphase cells. Therefore, expression silencing and overexpression of PolD1 might not simply mimic/exaggerate the interphase and mitotic states, respectively, and it would be an oversimplification to assume that merely the cytoplasmic ratio of PolD1 and γ TuRCs determines the level of the inhibition. The Golgi-associated PolD1 function could potentially be regulated at various levels, such as through the binding of PolD1 to γ TuRCs and the recruitment of PolD1 to the Golgi, both in interphase and in mitosis by as yet unidentified mechanisms.

We have now demonstrated the Golgi localization of PolD1 (Figure 5b–c of the revised manuscript). We performed *in vitro* reconstitution assays and demonstrated that PolD1 binds directly to γ TuRCs and inhibits their microtubule-nucleating activity (Figure 2 of the revised manuscript). As suggested, we examined the protein factors that are recognized to participate in microtubule nucleation and stabilization at the Golgi, including AKAP450, CDK5RAP2, and CLASPs. We found that the Golgi localization of these proteins was not affected by PolD1 overexpression, although the overexpression clearly inhibited Golgi-associated microtubule nucleation. The results have been included in the revised manuscript (Supplementary Figure 2b–d). Overall, the data presented in the manuscript strongly suggest that PolD1 controls Golgi-derived microtubule growth by inhibiting γ TuRC-mediated microtubule nucleation at the Golgi. Furthermore, our results do not support the sequestration of γ TuRCs in the cytosol.

The template model of microtubule nucleation by γ TuRCs was established in early studies by several groups (Keating and Borisy, *Nature Cell Biol.* 2000, 2:352–357; Wiese and Zheng, *Nat Cell Biol.* 2000, 2:358–364; Moritz et al., *Nature Cell Biol.* 2000, 2:365–370). More recently, Agard and colleagues have further demonstrated that the assembly of γ TuSCs into a ring structure and the closure of the ring permit the complexes to initiate microtubule nucleation (Kollman et al., *Nature* 2010, 466:879–882; *Nat Struct Mol Biol.* 2015, 22:132–137). Wieczorek et al. (*Nat*

Cell Biol. 2015, 17:907–916) have shown that the kinetics of template-mediated microtubule nucleation can be affected in a positive or negative manner by microtubule-associated proteins, such as XMAP215, TPX2, MCAK, and EB1. However, we have not detected any microtubule- or α/β -tubulin-binding activity of PolD1. Therefore, PolD1 is unlikely to act as a microtubule-associated protein and thereby affect nucleation kinetics and subsequent microtubule elongation and stabilization.

Minor concerns:

Pg. 3 line 10-11. Which other proteins were detected in the mass spectrometry? If the authors do not wish to release the full list, the peptide count for PolD1, cut-offs used to remove false positive results etc. should be included. If this list was previously published, authors should reference this.

Response:

We have added the mass spectrometry data of PolD1 identification and provided information such as cut-offs and sequence coverage (Supplementary Table 1). Currently, we are still in the process of identifying other proteins from the γ TuRC preparations and will publish the data when the work is completed.

Pg. 3 line 19 – pg. 4 line 2. These pull down experiments should also be performed with overexpressed PolD1 as many of the later experiments use this as a method to show PolD1 regulates GDMT levels. Does PolD1 overexpression disrupt the γ -TuRC?

Response:

In the manuscript, we have presented results showing the immunoprecipitation of overexpressed PolD1 and the coimmunoprecipitation of the γ TuRC subunits γ -tubulin and GCP 3–6 with the transiently expressed PolD1 (Figure 1c, d). Furthermore, we have shown that γ TuRC disassembly caused by RNAi-mediated depletion of GCP4 inhibited the coimmunoprecipitation of γ -tubulin with the overexpressed PolD1 (Figure 1c). These results clearly indicate that overexpressed PolD1 binds to intact γ TuRCs without disrupting them.

Pg. 4 line 2-3. Were other Pol δ subunits pulled down with transiently expressed PolD1? Control experiment should be included to show that the construct does not affect its known interactions.

Response:

We immunoprecipitated ectopically expressed PolD1 and immunoblotted the immunoprecipitates with an anti-PolD2 antibody; PolD2 is a scaffold subunit of Pol δ and it binds directly to PolD1. PolD2 was robustly detected in the immunoprecipitates of the expressed PolD1, and γ TuRC disassembly caused by GCP4 knockdown did not affect the coimmunoprecipitation of PolD2 with PolD1. These data have been

included in the revised manuscript (Figure 1c). The results indicate that the epitope tagging of PolD1 did not affect its interaction with PolD2. Previously, the functional Pol δ holoenzyme was reconstituted using epitope-tagged Pol δ subunits, including PolD1 (Zhou et al., PLoS One 2012, 7:e39156). Therefore, the epitope tags used do not affect PolD1 assembly into Pol δ .

Pg. 4 line 7-8 Figure 1. Quantification of reduced interactions? As a control, can the authors show that the γ -TuRC is actually disrupted by looking at other components, or is only GCP4 missing and/or are γ -TuSCs still intact?

Response:

As suggested by the reviewer, we have quantified the reduction of the interaction between PolD1 and γ -tubulin and included the data in the revised manuscript (Page 5, paragraph 2). GCP4 is widely accepted to be indispensable for γ TuRC assembly in animal cells, but it is not required for γ TuSC assembly (reviewed by Kollman et al., Nat Rev Mol Cell Biol. 2011, 12:709–721). Specifically, we and others have previously shown that RNAi-mediated suppression of GCP4 expression causes γ TuRC disassembly (Vérollet et al., J Cell Biol. 2006, 172:517–528; Choi et al., J Cell Biol. 2010, 191:1089–1095; Cota et al., J Cell Sci. 2017, 130:406–419). In our previous study (Choi et al., J Cell Biol. 2010, 191:1089–1095), γ TuRC disassembly was analyzed using two independent approaches, coimmunoprecipitation and sucrose-gradient centrifugation, and γ -tubulin and GCP 3–6 were all examined in the experiments. Therefore, we and others have clearly demonstrated that suppression of GCP4 expression disrupts γ TuRC assembly, but does not affect γ TuSC assembly.

Pg. 5 line 13-15. How do the IC₅₀s relate to native conditions? What is the stoichiometry of γ -TuRC to PolD1?

Response:

IC₅₀ values are a measure of inhibition potency. To relate the measured IC₅₀ value of PolD1 to physiological conditions, we estimated the cellular concentration of PolD1 based on the data on PolD1 abundance in mammalian cells and the size of mammalian cells (Page 15, paragraph 1), which revealed that the cytoplasmic concentration of PolD1 lies in a range that is close to the IC₅₀ value. Our finding suggests that PolD1 mediates a highly effective control of γ TuRC activities in the cytoplasm. We have added a discussion of the cytoplasmic concentration and the IC₅₀ value of PolD1 in the revised manuscript (Page 15, paragraph 1). However, we have not been able to determine the stoichiometry of PolD1 to the γ TuRCs because of technical challenges.

Pg. 6 line 8-9. Is GDMT nucleation fully inhibited, or is it slower? The centrosomal asters in GFP-PolD1 expressing cells look smaller and the MTs look shorter. Are GDMTs still absent and centrosomal MTs shorter or slower-growing over time? A time series (or a quantification thereof) would answer this question.

Response:

We made the following two observations: (1) Golgi-associated microtubule growth was not detected in PolD1-overexpressing cells even after a prolonged regrowth; and (2) the amount of Golgi-derived microtubules was drastically reduced in cells overexpressing PolD1. These and other observations presented in this manuscript together have led us to conclude that PolD1 overexpression inhibits Golgi-associated microtubule nucleation. As suggested, we have added the quantification data of Golgi-associated microtubule regrowth in the revised manuscript (Figure 3c). We observed that PolD1 overexpression slightly affected centrosome-based microtubule regrowth (Figure 3), as mentioned by the reviewer. However, PolD1 knockdown did not alter centrosome-based microtubule regrowth, whereas the knockdown promoted Golgi-associated regrowth (Figure 4). Therefore, PolD1 exerts its effect at the Golgi but not on centrosomes to control γ TuRC-dependent microtubule nucleation.

Pg. 7 line 19-20. 'low levels', compared to what? How much lower than other cells? Are other factors associated with GDMT nucleation/stabilization also altered in these cells? How do alterations in expression level affect localization? Is PolD1 present in the cytosol or at the Golgi in these low-expressing cells?

Response:

We observed that the PolD1 protein level varies in hTERT-RPE1 cells, and this agrees with the observations of others (Chea et al., Cell Cycle 2012, 11:2885–2895). We quantified the PolD1 levels and found that the PolD1 content of the lowest 5% cells was ~5-fold lower than that of the highest 5% cells. However, the PolD1 staining patterns of the cells that expressed PolD1 at distinct levels were highly similar. Therefore, the protein level variations do not noticeably affect PolD1 localization. We also examined the localization of three proteins involved in the nucleation and stabilization of Golgi-derived microtubules: CDK5RAP2, AKAP450, and CLASPs. Notably, PolD1 overexpression did not exert any effect on the Golgi localization of these proteins, and thus the variation of PolD1 levels did not markedly change the Golgi content of these proteins; these data have been included in the revised manuscript (Supplementary Figure 2b–d).

Pg. 8 line 17. Write out '2'.

Response:

We have written out “2” as “two” in the revised manuscript.

Pg. 9 line 15-16. Is Golgi orientation completed at all in PolD1 depleted cells eventually? Time series should be extended to determine this.

Response:

The depletion of PolD1 caused a delay in the Golgi reorientation process, and the reorientation was eventually completed, as shown in the time-lapse imaging results (Figure 7b and Supplementary Video 2).

Pg. 10 line 13-14. Is PolD1 found at the Golgi?

Response:

As stated above, we clearly detected PolD1 at the Golgi complex by immunoblotting isolated Golgi membranes and by immunostaining hTERT-RPE1 cells. The results have been added into the revised manuscript (Figure 5b, c).

Pg. 10 line 17. This is cell-type specific and should be mentioned as such.

Response:

We have revised the sentence accordingly and stated the cell type as “hTERT-RPE1” in the revised manuscript (Page 7, paragraph 2).

Pg. 11 line 2. Write out ‘2’.

Response:

We have changed “2” to “two” in the revised manuscript.

All figures showing fluorescence imaging: How are images shown here? Are they single frames, Z-stack projections, etc.? How were they processed; were gamma- or other non-linear adjustments made anywhere?

Response:

We have added the image processing information in the Methods section (Page 25, paragraph 3).

Figure 1.

1d. Were equal amounts of protein loaded after the IP? The FLAG-PolD1 levels seem variable in the image on the right (IP FLAG), especially the S605del band, which looks smaller than the others.

1c-d should have a control lane where an anti-FLAG antibody was used to show that when FLAG is expressed (in control (1c) or vector (1d)) but is not linked to PolD1, the interaction with γ -TuRC components is absent. The authors should specify that for the FLAG-PolD1 WB an antibody against PolD1 was used, not against FLAG.

Response:

Figure 1d: An equal fraction of the immunoprecipitates was used for SDS-PAGE and immunoblotting. Although the level of the immunoprecipitated S605del is slightly lower than those of wild-type and D515V proteins, it does not affect our conclusion.

Figure 1c–d: We have included a control in which the FLAG vector was transfected instead of the FLAG-PolD1 constructs, and we have labeled the FLAG vector-transfected samples as “Vector” in the figures. We detected immunoprecipitated FLAG-PolD1 in anti-FLAG immunoblots and not anti-PolD1 immunoblots, and we have indicated the anti-FLAG immunoblotting in the figure legend. Moreover, the expressed FLAG peptide is undetectable in immunoblotting because it is extremely small.

Figure 2.

2a. see comment above on 1c-d, FLAG control immunoblotting should be shown.

2b. The y-axis label should read γ -TuRCs with microtubules (%) or something to that effect. Were γ -TuRCs labelled in any way to determine this percentage? A (supplementary) image should be included of a control experiment without isolated γ -TuRCs but with CDK5RAP2 to show that under the used conditions no MT nucleation can occur without γ -TuRCs.

Response:

Figure 2a: In the *in vitro* assay shown in Figure 2a, the purified FLAG-PolD1 protein was tested for binding to the γ TuRCs. In the control, the γ TuRCs were incubated without FLAG-PolD1 and then subjected to anti-FLAG immunoprecipitation. This control is described in the figure legend. In the revised manuscript, we have added the anti-FLAG immunoblot to show the precipitated FLAG-PolD1 (Figure 2a).

Figure 2b: It will be inaccurate to label the y-axis as “ γ -TuRCs with microtubules (%)” We believe that even in the absence of PolD1, only a fraction of the γ TuRCs nucleated microtubules. We are unable to label isolated γ TuRCs, because chemical labeling will very likely affect their nucleation activity. To more accurately reflect the number of nucleated microtubules relative to that in the absence of PolD1, we have amended the y-axis label to “Microtubule numbers (%)” In a previous study (Choi et al., J Cell Biol. 2010, 191:1089–1095), we showed that CDK5RAP2 does not induce microtubule nucleation in the absence of γ TuRCs. As suggested by the reviewer, we performed the control nucleation experiment in which the CDK5RAP2 protein was included but the γ TuRCs were not. The results agreed with those in our previous report. We have included the data in Figure 2b.

Figure 4. 4a. What does ‘Golgi compactness was quantified as the proportion of TGN46-occupied space in the Golgi area’ mean? The Golgi area was outlined based on the TGN46 staining, yet this was also used to measure the compactness within that area?

4a. Why is γ -tubulin immunostaining shown? This panel is not referenced anywhere in the paper.

4b. Quantification of Golgi positioning should be added.

4c. Statistics should be added to the quantification graphs.

Are all scale bars 10µm? In 4b the scale bar looks to be shorter than that based on the nuclear sizes compared between the three data panels.

Response:

We have added a paragraph in the Methods section to describe how we measured Golgi compactness (Page 24, paragraph 3). Briefly, Golgi compactness is the proportion of the area occupied by the Golgi stacks (TGN46 signal) within the total area of the Golgi contour. This method has also been used by others for quantifying Golgi compactness (Vinogradova et al., Mol Biol Cell. 2012, 23:820–833).

Figure 4a (Figure 6a of the revised manuscript): We have added a statement to indicate that γ -tubulin was also stained for labeling centrosomes (Page 11, paragraph 1).

Figure 4b (Figure 6b of the revised manuscript): We have revised the result statement and indicated that Golgi positioning relative to the centrosomes was not affected by the overexpression or expression silencing of PolD1 (Page 11, paragraph 1). We found that the centrosomes were still closely associated with the Golgi complex in every examined cell. Quantification of the relative positioning is unnecessary because dissociation of the centrosomes from the Golgi complexes was not observed.

Figure 4c (Figure 6c of the revised manuscript): The statistics has been included in the figure legend.

We checked the scale bars in original Figure 4 (Figure 6 of the revised manuscript), and we confirm that they all represent 10 µm. Moreover, the scale bars are all in similar proportions to the sizes of the cell nuclei in the different figure panels, which support the consistency in image scale.

Figure 5. 5a-c. Statistics should be added.

5a. Why is the γ -tubulin panel included? It is not referenced anywhere in the manuscript.

Response:

The statistics has been included in the figure legend (Figure 7 of the revised manuscript). In Figure 5a (Figure 7a of the revised manuscript), γ -tubulin was stained to label centrosomes, which typically associate with the Golgi complex. We have added a statement about the labeling in the manuscript (Page 12, paragraph 2).

Supplementary Figure 2. The FLAG-PolD1 image shows some signal in the three cells on the right. Are these cells transfected and weakly expressing FLAG-PolD1? If so, authors should also include a non-transfected cell to show no effect on γ -tubulin levels at the centrosome.

Response:

We cannot exclude the possibility that the very weakly stained cells were transfected and expressed FLAG-PolD1 at extremely low levels. Therefore, we reperformed the experiments, but now used GFP and GFP-PolD1 constructs and compared the centrosomal staining of γ -tubulin in GFP- and GFP-PolD1-transfected cells. We have replaced the images with the new data (Supplementary Figure 2a of the revised manuscript).

Supplementary Figure 3 c. Quantification of GDMT levels should be added.

Response:

We have added the quantification of Golgi-based microtubules in Supplementary Figure 3c.

Methods.

Mass spectrometry section is missing.

Cell culture and immunofluorescence microscopy. 6th line of the section, a '/' is missing between penicillin and streptomycin.

Response:

We have added the mass spectrometry methods in the Methods section (Page 21, paragraph 3). We have also added “/” between penicillin and streptomycin.

Reviewer #2:

General comments:

In this manuscript, Shen et al. extend their previous work on isolating the gamma-tubulin ring complex (γ TuRC) and found PolD1, the catalytic subunit of the DNA polymerase δ , co-purified with γ TuRCs. Through co-immunoprecipitation and western blotting analysis, they confirmed that PolD1 is in a complex with intact γ TuRCs and this association requires neither its DNA polymerase nor exonuclease activities. The authors then demonstrated that, in the presence of recombinant PolD1 protein, Cdk5Rap2-stimulated microtubule nucleation by γ TuRCs was attenuated in vitro, which suggests that PolD1 is a γ TuRC inhibitor. PolD1 overexpression impaired microtubule nucleation at the Golgi and rendered the Golgi more compact. Conversely, PolD1 knockdown increased microtubule nucleation at the Golgi and caused the Golgi to expand. Consequently, PolD1 RNAi resulted in defects in Golgi reassembly after nocodazole washout and in Golgi reorientation in a scratch wound assay. Based on these results the authors concluded that PolD1 negatively regulates Golgi-derived microtubule nucleation by inhibiting γ TuRCs.

If correct, the findings of a Golgi-specific γ TuRC inhibitor would be novel. However, the authors need to demonstrate that the effects are specific and direct. The major concern is that, the conclusions rely on overexpression and knockdown of PolD1,

which is a catalytic subunit of the DNA polymerase δ . Given the essential functions of DNA polymerases, it is to be expected that these manipulations will induce adverse effects to the cells. The authors need to experimentally dissect the different functions of PolD1 and use rigorous controls to rule out indirect effects.

A second and more profound issue is the lack of mechanistic insights into how PolD1 might regulate γ TuRC at the Golgi. Conceptually, it is difficult to reconcile the nuclear localization and the well-established function of PolD1 in DNA replication/repair with a Golgi-specific activity. The authors co-immunoprecipitated PolD1 with γ TuRCs from detergent lysates, and it is not clear whether PolD1 was released from the nucleus during lysis. At a minimum, the authors need to localize endogenous PolD1 to Golgi membranes by immunofluorescence or EM, as well as subcellular fractionation (without detergent). Without this evidence, it is likely that co-purification and co-immunoprecipitation of PolD1 with γ TuRCs is induced during detergent lysis when nuclear PolD1 is released into the cytosol.

On balance, the authors' findings are interesting but too preliminary to conclude that PolD1 regulates microtubule nucleation at the Golgi. In its present form, the manuscript is rather descriptive and correlative. There are significant holes in the story from the perspectives of both biochemistry and cell biology that need to be addressed before the manuscript is suitable for publication.

Response:

The reviewer's comments encouraged us to further explore the mechanism of PolD1 action in the organization of Golgi-derived microtubules. We have now obtained the following experimental data to further support our conclusion that PolD1 acts directly on γ TuRCs to control Golgi-associated microtubule growth. First, we performed subcellular fractionation and determined that a substantial amount (~63%) of PolD1 localizes in the cytoplasm (Figure 5a). Second, we demonstrated the Golgi localization of PolD1 by using two approaches: (1) we isolated Golgi membranes from hTERT-RPE1 cells by using a well-established protocol and detected PolD1 in the isolated membranes by immunoblotting; and (2) we visualized PolD1 at the Golgi through immunofluorescence staining of hTERT-RPE1 cells (Figure 5b, c). Third, to specifically address whether the nuclear localization of PolD1 is required for the γ TuRC-associated function, we created a PolD1 construct lacking the nuclear localization signal. Although this PolD1 construct did not exhibit nuclear localization, it retained the γ TuRC-binding activity and the ability to inhibit Golgi-associated microtubule nucleation (Figure 5d, e). These analyses have dissected the nuclear and cytoplasmic functions of PolD1 and the results have unequivocally shown that the γ TuRC-binding and -inhibiting functions of PolD1 are independent from PolD1 localization and functions in the nucleus. Collectively, our new data indicate that the γ TuRC inhibition by PolD1 occurs at the Golgi complex.

In our previously submitted manuscript, we presented data from *in vitro* reconstitution assays to show the direct interaction between PolD1 and γ TuRCs and the inhibition of CDK5RAP2-stimulated γ TuRCs by PolD1 (Figure 2a, b). In the revised manuscript, we have added data from *in vitro* assays of γ TuRC-induced microtubule nucleation in the absence of CDK5RAP2; our results show that PolD1 inhibited the microtubule-nucleating activity of γ TuRCs that were free of the

CDK5RAP2 protein (Figure 2c). Together, these data clearly indicate that PolD1 exerts a direct inhibitory effect on γ TuRCs, the microtubule nucleators that are indispensable for Golgi-associated microtubule growth.

We also examined cell cycle progression in PolD1-depleted cells, which revealed that the RNAi-mediated depletion caused only a slight increase (by ~10%) of S-phase cells (Page 8, paragraph 2). This agrees with the observations of others (Bermudez et al., J Biol Chem. 2011, 286:28963–28977; Tumini et al., Sci Rep. 2016, 6:38873). The efficiency of PolD1 knockdowns was ~90% (Supplementary Figure 3a), and most of the PolD1-depleted cells exhibited the presented phenotypes of Golgi-associated microtubule growth. Therefore, the observed minor effect of PolD1 knockdown on cell cycle progression cannot account for the large population of cells showing the Golgi-associated phenotypes. Moreover, a previous study reported that the activity of Golgi-associated microtubule growth in hTERT-RPE1 cells does not vary substantially when the cells progress from G1 to G2 (Maia et al., Cytoskeleton 2013, 70:32–43). Therefore, the Golgi-associated phenotypes observed in the PolD1-depleted cells are unlikely to be caused by a change in cell cycle progression.

Taken together, our analyses have unambiguously dissected the nuclear and cytoplasmic PolD1 functions and excluded the possibility that the nuclear function of PolD1 is involved in the control of γ TuRC-mediated microtubule nucleation at the Golgi. Furthermore, we have revealed the Golgi localization of PolD1 and strengthened the conclusion that PolD1 directly inhibits γ TuRCs. The findings presented in the revised manuscript provide key insights into the mechanism by which PolD1 controls Golgi-associated microtubule growth.

Specific comments

Major:

1. Figure 2b, in vitro microtubule nucleation assay:

The authors recognized that co-IP of PolD1 with γ TuRCs does not indicate their direct interaction and thus sought to reconstitute its direct inhibition on γ TuRCs in vitro using purified components. Unfortunately, this set of experiments lack essential controls and are unconvincing. As the authors noted, the reaction actually contains a Cdk5Rap2 domain that binds and stimulates γ TuRCs. PolD1 reduced this stimulation, which could be due to a direct inhibition on γ TuRCs as the authors proposed, or caused by the added PolD1 that sequesters/competes the Cdk5Rap2 domain and/or interferes its activity to stimulate γ TuRCs. To show a direct inhibition of γ TuRCs, the authors have to only use purified γ TuRCs, tubulins and PolD1. Furthermore, they need to dissect whether PolD1 interacts with the Cdk5Rap2 domain or interferes its binding to γ TuRCs. The reaction should also include specificity controls such as a same epitope-tagged γ TuRC binding-deficient mutant of PolD1 or a catalytic subunit of other DNA polymerases. In addition, since the polymerase-deficient and exonuclease-deficient PolD1 mutants interact with γ TuRCs, do they also suppress nucleation?

Response:

As suggested, we performed the *in vitro* microtubule nucleation assay by using isolated γ TuRCs that were free of the CDK5RAP2 domain protein. In the absence of the CDK5RAP2 protein, the isolated γ TuRCs exhibited weak microtubule-nucleating activity (Choi et al., J Cell Biol. 2010, 191:1089–1095), and this activity was strongly inhibited when the PolD1 protein was added (Figure 2c). We also tested the polymerase-deficient and exonuclease-deficient PolD1 mutants (S605del and D515V) and found that they showed similar γ TuRC-inhibiting activities as the wild-type protein (Figure 2c). These data together with the data on the direct binding of PolD1 to γ TuRCs (Figure 2a) clearly indicate that PolD1 acts directly on γ TuRCs to inhibit their microtubule-nucleating activity. Furthermore, our results demonstrate that PolD1 binds to and inhibits γ TuRCs regardless of the presence of CDK5RAP2. Therefore, we consider the characterization of the potential direct interaction between PolD1 and CDK5RAP2 to fall outside the scope of this study, which is focused on the direct action of PolD1 on γ TuRCs.

We agree that the interpretation of the observed γ TuRC inhibition by PolD1 in the *in vitro* assays of microtubule nucleation would benefit from the inclusion of specificity controls. Eukaryotic DNA polymerases constitute an ever-expanding family of proteins containing at least 15 members, some of which are single polypeptides whereas others are multisubunit enzymes (Hubscher et al., Annu Rev Biochem. 2002, 71:133–163). Moreover, the structure of the catalytic subunits varies among the DNA polymerases; for example, DNA polymerase α , a tetrameric enzyme involved in chromosome replication, lacks the 3'→5' exonuclease activity. Therefore, each DNA polymerase presents unique structural and functional properties. Although determining whether the γ TuRC-inhibiting function is unique to PolD1 among the DNA polymerases would be interesting, it is beyond the scope of this study. To focus our study on PolD1 and its related Pol δ proteins, we tested PolD2—a Pol δ core subunit that binds directly to PolD1—in the nucleation assays, which revealed that PolD2 exerted no effect on γ TuRC-mediated microtubule nucleation (Figure 2c). We have now included these data in the manuscript to strengthen the specificity of the observed γ TuRC inhibition by PolD1.

2. Figure 2c, PolD1 overexpression:

As mentioned above, the distribution and localization of endogenous PolD1 needs to be determined. How much PolD1 is localized to the nucleus, the cytoplasm and the Golgi (if any) respectively? The authors can access the relative ratio by immunofluorescence and subcellular (nuclear/cytosol/membrane) fractionations. Also, how does this ratio change when PolD1 is overexpressed or down-regulated? This information is needed to interpret their data. In Figure 2c, for example, overexpression of GFP-PolD1, both wildtype and two mutants, causes defects in microtubule nucleation at the Golgi complex. However, the expression levels of control GFP vs GFP-PolD1 are not presented. In addition, control GFP localizes predominantly to the nucleus, which is not a proper control for overexpressed PolD1 that has much higher cytosolic localization. Again the question is one of specificity.

Response:

We performed subcellular fractionation and immunoblotting to examine the nucleocytoplasmic distribution of endogenous and overexpressed PolD1, and the

results have been included in the manuscript. We found that the nuclear and cytoplasmic fractions contained ~37% and ~63% of endogenous PolD1, respectively (Figure 5a). Similarly, overexpressed PolD1 (i.e., GFP-PolD1) was distributed almost equally between the nuclear and cytoplasmic fractions (Supplementary Figure 5). In cells transfected with *polD1*-targeting siRNAs, the PolD1 level was reduced by ~90% (Supplementary Figure 3a), and therefore the residual PolD1 signal was too low for analyzing PolD1 distribution.

In the experiments in which GFP and GFP-PolD1 or the PolD1 mutants were expressed, the cells that expressed these proteins at similar levels were selected for analyzing the effects of the expression on microtubule regrowth. We have added a statement regarding this in the legends for Figures 3 and 5. In these immunostaining experiments, the cells were fixed in 4% paraformaldehyde plus 0.5% Triton X-100; under this condition, most of the cytosolic GFP was removed and thus the GFP imaging signal appeared predominantly in the nuclei. To verify the nucleocytoplasmic distribution of GFP, we performed subcellular fractionation and found that most of the GFP was present in the cytoplasmic fraction (Supplementary Figure 5). The cytoplasmic localization of GFP was corroborated by the results obtained when we performed imaging of GFP-transfected cells that were fixed in 4% paraformaldehyde in a detergent-free buffer. Therefore, GFP does not affect the nucleocytoplasmic distribution of GFP-PolD1. Our findings strongly support the conclusion that the observed effects are specifically due to the ectopically expressed PolD1.

3. Figures 3-5, PolD1 knockdown:

Most phenotypic and functional readouts heavily rely on PolD1 knockdown. The authors need to dissect whether the phenotypes were due to a direct PolD1 function at the Golgi or due to indirect secondary defects caused by inhibition of DNA replication. One possibility is that PolD1-RNAi cells may be enriched at a certain cell cycle stage when microtubule nucleation activity is coincidentally up-regulated. To address this concern, in addition to showing a part of PolD1 is localized to the Golgi, the authors should specifically inhibit cytoplasmic/Golgi-localized pool of PolD1 and investigate its outcomes. For instance, microinjections of PolD1 protein (since it works in vitro) and/or of an inhibitory antibody against PolD1 (if available) into the cytoplasm, or using a Golgi-targeting construct, etc. Alternatively and perhaps easier, and AID0PolD1 would at least allow measuring acute effects of PolD1 loss. In addition, can expression of RNAi-resistant forms of PolD1 wildtype and two mutants rescue the phenotypes?

Response:

As stated above, we have now provided evidence to show the Golgi localization of PolD1 (Figure 5b, c). Moreover, we examined cell cycle progression in the PolD1-depleted cells by performing flow cytometry, which revealed that the depletion of PolD1 caused only a slight increase (by ~10%) of the S-phase population (Page 8, paragraph 2). This cell cycle effect agrees with the observations of others (Bermudez et al., J Biol Chem. 2011, 286:28963–28977; Tumini et al., Sci Rep. 2016, 6:38873). Therefore, this small increase of S-phase cells cannot account for the large PolD1-depleted population (~90% efficiency of PolD1 knockdowns) showing the phenotypes of Golgi-associated microtubule growth. As mentioned earlier, the level

of Golgi-associated microtubule nucleation in hTERT-RPE1 cells does not change markedly during the cell cycle progression from G1 to G2 (Maia et al., Cytoskeleton 2013, 70:32–43). Together, these findings exclude the possibility that the Golgi-associated phenotypes observed in the PolD1-depleted cells are due to the increase of S-phase cells.

In this revised manuscript, we have included results showing that a substantial amount of PolD1 is present in the cytoplasm (Figure 5a). To specifically investigate whether the nuclear localization of PolD1 is required for the γ TuRC-associated function, we created a truncated PolD1 construct that lacks the nuclear localization signal. This truncated protein was not localized in the nucleus, but it retained the γ TuRC-binding activity and the ability to inhibit Golgi-associated microtubule nucleation (Figure 5d, e). Together with other data presented in the manuscript, these results strongly support the conclusion that PolD1 acts at the Golgi to inhibit γ TuRC-mediated microtubule nucleation.

As suggested by the reviewer, we performed the rescue experiments in which the expression of the endogenous protein was silenced by transfecting a siRNA targeting an untranslated region of *pold1* (PolD1 RNAi-3, Supplementary Figure 3a). The ectopic expression of PolD1 or its D515V and S605del mutants in these endogenous-PolD1-depleted cells substantially reduced Golgi-associated microtubule nucleation (Supplementary Figure 4). These results indicate that the cellular phenotypes observed after transfection of *pold1*-targeting siRNAs resulted specifically from the depletion of PolD1.

4. Supplementary Figures 4:

What percentage of the cells have low levels of PolD1? If this occurs at a high frequency, one cannot tell if the phenotypes are due to loss-of-function by RNAi or inherent heterogeneity within a cell population. Based on the reference cited by the authors (Ref. 27), the levels of PolD1 almost double in G2 phase compared to G1. Can the authors synchronize cells and compare the phenotypes between the different cell cycle stages? Are the results consistent with observed phenotypes?

Response:

We quantified the PolD1 levels in interphase hTERT-RPE1 cells and found that ~5% of the cells expressed PolD1 at considerably diminished levels. Therefore, only a small hTERT-RPE1 population contains PolD1 at low levels. As suggested by Lee and colleagues (Chea et al., Cell Cycle 2012, 11:2885–2895), these cells were likely not undergoing active DNA replication or repair. In the experiments involving RNAi-mediated silencing of PolD1 expression, most of the cells were transfected with the *pold1*-targeting siRNAs, and the knockdown efficiencies were ~90% (Supplementary Figure 3a). In the siRNA-transfected cells, we detected almost no PolD1 immunofluorescence (Figure 4 and Supplementary Figure 3b). These results exclude the possibility that the phenotypes observed in the siRNA-transfected cells are due to the inherent heterogeneity of PolD1 expression in hTERT-RPE1 cells. Thus, our data clearly indicate that the observed phenotypes are specific effects produced by the loss of PolD1.

Golgi-associated microtubule-nucleating activities were quantified at various cell cycle stages and compared by Kaverina and colleagues (Maia et al., Cytoskeleton 2013, 70:32–43), and the results showed that the measured activities were highly similar between G1 and G2 of hTERT-RPE1 cells. Therefore, the close-to-1-fold increase of PolD1 expression at G2 (Chea et al., Cell Cycle 2012, 11:2885–2895) does not necessarily lead to a substantial reduction of γ TuRC-mediated microtubule nucleation at the Golgi. The Golgi-associated function of PolD1 is likely regulated at various levels, such as through the binding of PolD1 to γ TuRCs and the nucleocytoplasmic localization and the Golgi recruitment of PolD1, by as yet unidentified mechanisms.

Minor:

1. Figure 1a: Do other subunits (PolD2, PolD3, PolD4) of the DNA polymerase δ also co-IP with γ TuRCs? In other words, is the interaction specific to PolD1?

Response:

In the mass spectrometry analyses performed on the isolated γ TuRCs, PolD1 was identified, but no other Pol δ subunit was found. Similarly, PolD2 was not detected when we immunoblotted the immunoprecipitates of γ TuRCs. Thus, it is highly likely that PolD1 that is not bound to other Pol δ subunits associates with γ TuRCs in the cytoplasm.

2. The following statements (underlined) are an over-interpretation of the data, and have not been convincingly shown in the current manuscript and should be modified.

The second last paragraph line 10:

*We have shown that PolD1 physically associates with γ TuRCs in the cytoplasm
-> The authors used whole cell lysate for the co-IP, which includes nucleoplasm where presumably the majority of PolD1 is present. They did not present any data demonstrating PolD1 is recruited from the cytoplasm to the Golgi membranes either.*

The second last paragraph line 14:

*This action of PolD1 occurs at the Golgi complex
-> Supporting data are not provided.*

Response:

In the revised manuscript, we have provided additional new evidence to show the cytoplasmic localization of PolD1. (1) The results of subcellular fractionation showed that the nuclear and cytoplasmic fractions contained ~37% and ~63% of PolD1, respectively; (2) both PolD1 and γ -tubulin were detected through immunoblotting in the isolated Golgi membranes; and (3) PolD1 was visualized by means of immunofluorescence staining at the Golgi complex (Figure 5a–c). Therefore, a substantial amount of PolD1 exists in the cytoplasm, and a fraction of the cytoplasmic PolD1 is attached to the Golgi complex. Moreover, whether γ TuRCs are present in the nucleus is unknown, although a small fraction of γ -tubulin was reported

to exist in the nucleus (Andersen et al., Curr Biol. 2002, 12:1–11). Our new data strongly support the conclusion that PolD1 interacts with γ TuRCs in the cytoplasm and inhibits γ TuRC-mediated microtubule nucleation at the Golgi complex.

Reviewer #3:

Interesting and well-written manuscript describes a novel and unexpected role of the catalytic subunit of main replicative DNA polymerase delta in microtubule dynamics. POLD1 directly binds to gammaTuRCs and inhibits microtubule nucleation. Known enzymatic activities of pol delta, exonuclease and polymerase, are apparently not required for the association. Depletion of POLD1 specifically enhances these processes at the Golgi, while overexpression inhibits them, altering its organization and cell behavior. The evidence presented by the authors is convincing, though some of the results are supported only by images with no quantitative estimates of the level of effects (Figs. 2 and 3).

Response:

We thank the reviewer for the positive comments. As suggested, we quantified the Golgi-based microtubule regrowth shown in Figures 2 and 3 (Figures 3 and 4 of the revised manuscript), and we have included the quantification data in the revised manuscript (Figures 3c and 4). We have also added the quantification data obtained from other microtubule regrowth experiments (Figure 5e and Supplementary Figure 3c).

Several things need to be clarified.

Pol delta is not the most abundant protein in the cell, while components of microtubules are. It will be good to discuss the stoichiometry of the inhibitory processes.

There is some confusion with RNAi inhibition experiment (sFig 3 and Fig. 3). POLD1 is vitally required for cell growth. Images of cells show no pol delta detectable by immunostaining in the nuclei. The authors should describe what else happens with these cell lines in addition to changes in Golgi.

It should be better explained what is the mechanism of specificity of pol delta to gammaTuRCs in Golgi. Ideas how free catalytic subunit of the tight pol delta complex appears in cells should be presented.

The most harmful weakness is a very limited discussion of biological meaning and significance of the discovered effect.

Response:

We estimated the cellular concentration of PolD1 based on the data on PolD1 abundance in mammalian cells and the size of mammalian cells (Page 15, paragraph 1), and found that the cytoplasmic concentration of PolD1 is in a range that is close to the IC₅₀ value of γ TuRC inhibition by PolD1. However, because of technical challenges, we have not been able to determine the PolD1 stoichiometry required for inhibiting γ TuRCs. We have added a discussion on the cytoplasmic concentration and the IC₅₀ value of PolD1 in the manuscript (Page 15, paragraph 1). PolD1 is also likely

to be enriched at specific cytoplasmic sites, such as the Golgi complex. Accordingly, we detected PolD1 in isolated Golgi membranes through immunoblotting and visualized the Golgi localization of PolD1 by means of cell immunostaining; these results are included in the revised manuscript (Figure 5b, c). Moreover, we have presented data to show that changes in the cytoplasmic PolD1 level affect the activity of Golgi-associated microtubule nucleation.

We found that RNAi-mediated depletion of PolD1 caused only a slight increase (by ~10%) of S-phase cells. This agrees with the observations of others (Bermudez et al., J Biol Chem. 2011, 286:28963–28977; Tumini et al., Sci Rep. 2016, 6:38873). We have now described this knockdown effect on the cell cycle in the manuscript (Page 8, paragraph 2).

From the isolated γ TuRCs, we identified PolD1 by using mass spectrometry, but did not find any other Pol δ subunits, which implies that the γ TuRC-associated PolD1 is not bound to other Pol δ subunits. In addition to the Pol δ heterotetramer, monomeric PolD1 and the PolD1-PolD2 heterodimer exist in mammalian cells (Goulian et al., J Biol Chem. 1990, 265:16402–16411). It is highly likely that the PolD1 that is not bound to other Pol δ subunits associates with γ TuRCs in the cytoplasm. We have now included these explanations in the Discussion section (Page 14, paragraph 2).

We are grateful for the suggestion about expanding the discussion of the biological meaning and significance; we have expanded the Discussion section and further addressed the biological meaning and significance of the findings presented in this manuscript (Page 14, paragraph 2; Page 15, paragraphs 1-3; Page 16, paragraphs 1-3; Page 17, paragraph 1).

Minor comments:

It is not clearly described, what was the source of isolated gammaTuRCs when pol delta was first detected.

Brief description of human cell lines and rationale for choosing the particular ones (two cancer lines, one immortalized normal eye epithelium) would be helpful.

Graphs in Fig. 4 occupy random positions. They should be labeled, so the relation to images will be clear.

Response:

γ TuRCs were isolated from HEK293T cells for mass spectrometry analyses and PolD1 was identified from the isolated samples. We have included this material source information in the revised manuscript (Page 20, paragraph 2).

We have now briefly described the rationale for using HEK293T and hTERT-RPE1 cell lines in the revised manuscript (Page 7, paragraph 2; Page 20, paragraph 2).

As suggested, we have rearranged the panel positions in Figure 4 (Figure 6 of the revised manuscript) to clarify the relationship between the quantification graphs and the micrographs.

Reviewers' Comments:

Reviewer #1:

Remarks to the Author:

The manuscript is significantly strengthened by the revision, and the major hypotheses have been substantiated. All my concerns have been addressed. One small remaining thing is to provide quantification of PolD1 intensity at the Golgi versus other cell localizations in the images provided in Fig. 5C,E.

Reviewer #2:

Remarks to the Author:

In this revised manuscript, Shen et al. included additional experiments in response to the concerns raised in the first review, in particular the Golgi-specific localization of PolD1 and its direct inhibition toward γ TuRCs in vitro.

The paper now convincingly shows that:

- PolD1 overexpression decreases microtubule nucleation density around the Golgi and renders the Golgi more compact.
- PolD1 knockdown increases microtubule nucleation density around the Golgi and causes Golgi expansion.
- PolD1 knockdown results in defects in Golgi reassembly after nocodazole washout and in Golgi reorientation in a scratch wound assay.

However, the link of these macroscopic phenotypes to the molecular player, i.e. PolD1 as a direct γ TuRC inhibitor at the Golgi, remains circumstantial.

To determine whether the nuclear protein PolD1 also moonlights as an inhibitor for γ TuRCs at the Golgi, the authors need to unambiguously establish that PolD1 is recruited to the Golgi membrane. To this end, the authors performed immunofluorescence, subcellular fractionations, and improved the microtubule nucleation assay. Several lines of evidence presented, however, remain inconsistent and lack proper controls, which leaves the results inconclusive. In particular, the following key results are confusing and, in some cases, contradict each other.

1) Localization of over-expressed PolD1

As discussed before, in Figure 3, for example, control GFP localizes predominantly to the nucleus, which is not an adequate control for overexpressed PolD1 that has a much greater cytosolic distribution. In addition, these immunofluorescence data contradict the subcellular fractionation results in supplementary Figure 5 where GFP is present almost exclusively in the cytosolic fraction. The authors explained that the nuclear localization of GFP by IF was due to the presence of triton during fixation, which caused the majority of cytosolic proteins to be washed out. This explanation is rather unsatisfactory, because (a) triton permeabilizes both nuclear membrane and plasma membranes, which should extract comparable amounts of proteins from both compartments (b) If their reasoning is correct, how can the authors explain that under the same condition only the cytosolic fraction of GFP, but not GFP-PolD1 was extracted?

2) Golgi membrane enrichment

Since the authors did not perform immuno-EM to localize PolD1 precisely, one would hope that the membrane fractionation data would be suggestive in terms of its association with the Golgi apparatus. Unfortunately, Figure 5b does not show any enrichment of Golgi proteins in their prep. There is no loading control described or any cytosolic and membrane markers other than the Golgi being used to evaluate membrane enrichment of their prep. To evaluate the enrichment of PolD1

on Golgi membranes, equal amounts of total proteins from fractions of the PNS, the partially enriched intermediate along fractionation procedures, and the final highly enriched Golgi membranes should be loaded and compared. Furthermore, the blots should be probed for markers of the cytosol, ER, mitochondria and PM to demonstrate the specific association of PolD1 with Golgi membranes.

3) In vitro microtubule nucleation assay

The authors now use purified γ TuRCs (without the Cdk5Rap2 TuNA domain) and purified PolD1, including wt, the polymerase-deficient and the exonuclease-deficient PolD1 mutants. All three bind to γ TuRCs by co-IP and inhibit γ TuRC-mediated microtubule nucleation in vitro. As a specificity control, the authors include another Pol δ subunit, PolD2, which does not attenuate γ TuRC-mediated microtubule nucleation. However, all three forms of PolD1 (wt, D515V, S605del) were expressed and purified from mammalian HEK293T cells but the negative control PolD2 was expressed and purified from bacteria. The authors should express and purify this crucial control protein also from HEK293T cells to rule out the possibility that a co-purified unspecific protein or factor (that is absent in bacteria) accounts for the observed activity.

In addition to the above specific points, the author should address their statement "Golgi-derived microtubules are required for Golgi assembly and maintenance" (in abstract and text) more carefully. In the recently published Ref. 6 (which the corresponding author of this manuscript co-authored), Akhmanova's group demonstrated that a compact Golgi can be assembled in the absence of both centrosomal and Golgi microtubules. So how are these data reconciled with the phenotypes that authors describe here? The authors should examine Golgi-derived microtubules using a cleaner system where centrosomal microtubules are depleted (such as knockout or centrinone treatment), as the Golgi-derived microtubules are so closely placed to the centrosomes, which very likely confounds authors' quantitation and interpretation.

In summary, I appreciate the authors' efforts in performing many experiments during revision, but given the inconsistency and data quality presented in the revised manuscript, as well as lack of mechanistic insights, unfortunately I cannot recommend its publication in Nature Communications. However, I encourage the authors to further clarify these issues and submit it to a more specialized journal.

Reviewer #3:

Remarks to the Author:

The revised manuscript with additional experimental data makes the work much more convincing than that previous version. Novel function of POLD1 is exciting link between microtubules and DNA metabolism,

Point-by-point responses to reviewers' comments:

Reviewer #1:

The manuscript is significantly strengthened by the revision, and the major hypotheses have been substantiated. All my concerns have been addressed. One small remaining thing is to provide quantification of PolD1 intensity at the Golgi versus other cell localizations in the images provided in Fig. 5C,E.

Response:

In the experiments presented in Figure 5c (Figure 5b in the revised manuscript), cells were extracted in a saponin-containing buffer before fixation and immunostaining in order to remove cytosolic proteins. Consequently, the Golgi and nuclear patterns are the most prominent signals of PolD1 staining. We have now determined the PolD1 signal intensities at the Golgi relative to that in the nucleus, and these quantification data are provided in the legend of Figure 5b in the revised manuscript. In the experiments shown in Figure 5e, the truncated PolD1 construct 247–955, which lacks the nuclear-localization signal, was used, and this construct did not exhibit any detectable nuclear localization. Thus, we believe that the amount of 247–955 at the Golgi relative to that in the nucleus or other subcellular locations is likely to differ from the relative amounts of the full-length protein under physiological conditions. Therefore, we did not quantify the subcellular signals of 247–955.

Reviewer #2:

In this revised manuscript, Shen et al. included additional experiments in response to the concerns raised in the first review, in particular the Golgi-specific localization of PolD1 and its direct inhibition toward γ TuRCs in vitro.

The paper now convincingly shows that:

- PolD1 overexpression decreases microtubule nucleation density around the Golgi and renders the Golgi more compact.*
- PolD1 knockdown increases microtubule nucleation density around the Golgi and causes Golgi expansion.*
- PolD1 knockdown results in defects in Golgi reassembly after nocodazole washout and in Golgi reorientation in a scratch wound assay.*

However, the link of these macroscopic phenotypes to the molecular player, i.e. PolD1 as a direct γ TuRC inhibitor at the Golgi, remains circumstantial.

To determine whether the nuclear protein PolD1 also moonlights as an inhibitor for γ TuRCs at the Golgi, the authors need to unambiguously establish that PolD1 is recruited to the Golgi membrane. To this end, the authors performed immunofluorescence, subcellular fractionations, and improved the microtubule nucleation assay. Several lines of evidence presented, however, remain inconsistent and lack proper controls, which leaves the results inconclusive. In particular, the following key results are confusing and, in some cases, contradict each other.

Response:

Our manuscript includes the following key findings, in addition to those summarized by the reviewer: (1) PolD1 binds directly to γ TuRCs and inhibits γ TuRC-mediated microtubule nucleation, and (2) PolD1 associates with the Golgi complex. In this revised manuscript, we have strengthened these points by performing the following experiments. First, we prepared the PolD2 protein from HEK293T cells under conditions identical to those used for PolD1. By using this PolD2 protein as a control in the microtubule nucleation assay, we eliminated the possibility that nonspecific contaminants (if any) in the PolD1 preparations cause the γ TuRC inhibition. Second, bolstering our results of PolD1 immunostaining at the Golgi, we showed the co-isolation of PolD1 with the Golgi by means of equilibrium sucrose-gradient centrifugation. The data from the immunostaining and the biochemical analysis have clearly shown the localization of PolD1 at the Golgi complex. Collectively, our findings have conclusively demonstrated that PolD1 is a γ TuRC inhibitor that controls microtubule nucleation at the Golgi.

The reviewer's comment that "Several lines of evidence presented, however, remain inconsistent and lack proper controls, which leaves the results inconclusive" is a very general statement without any indication of which experiments lack proper controls. We also consider this statement to be biased to a certain extent, as we explain below in our responses to the specific comments. In the following sections, we carefully address all of the specific issues raised by this reviewer.

1) Localization of over-expressed PolD1

As discussed before, in Figure 3, for example, control GFP localizes predominantly to the nucleus, which is not an adequate control for overexpressed PolD1 that has a much greater cytosolic distribution. In addition, these immunofluorescence data contradict the subcellular fractionation results in supplementary Figure 5 where GFP is present almost exclusively in the cytosolic fraction. The authors explained that the nuclear localization of GFP by IF was due to the presence of triton during fixation, which caused the majority of cytosolic proteins to be washed out. This explanation is rather unsatisfactory, because (a) triton permeabilizes both nuclear membrane and plasma membranes, which should extract comparable amounts of proteins from both compartments (b) If their reasoning is correct, how can the authors explain that under the same condition only the cytosolic fraction of GFP, but not GFP-PolD1 was extracted?

Response:

The reviewer's concern regarding the GFP staining pattern was already addressed in the previous revision of this manuscript. Here, we further elaborate on this point and explain and emphasize that GFP-only is an appropriate control in our microtubule regrowth experiments involving the GFP fusion constructs of PolD1. In microtubule regrowth assays, cell treatment with methanol at -20°C is a commonly used fixation method for visualization of Golgi-based regrowth (Grimaldi, Fomicheva, and Kaverina, *Methods Cell Biol.*, 2013, 118: 401–415). However, methanol fixation is incompatible with the visualization of GFP fluorescence (Nybo,

Biotechniques, 2012, 52: 359–360). Therefore, in the experiments involving GFP and GFP fusion constructs (e.g., assays shown in Figure 3), we fixed the cells with 4% paraformaldehyde in a buffer containing 0.5% Triton X-100 in order to visualize both GFP signals and Golgi-associated microtubules. This procedure of concurrent extraction and fixation extracted cytosolic soluble proteins, but the nuclear signals were retained. Under this condition, the GFP proteins, especially GFP alone, exhibited clear nuclear staining signals.

Fixation and extraction procedures are widely recognized to exert diverse effects on the immunofluorescence staining patterns of soluble proteins in cultured cells (Melan and Sluder, J Cell Sci. 1992, 101, 731–743). In our previous revised manuscript, we provided subcellular fractionation data to show that most of the expressed GFP existed in the cytoplasmic fraction (Supplementary Figure 5). Furthermore, we stated in our responses last time that the cytoplasmic localization of GFP was corroborated by the fluorescence imaging results obtained from GFP-transfected cells that were fixed in 4% paraformaldehyde in a detergent-free buffer. It is disappointing that these findings were overlooked by the reviewer. We would also like to point out that the cytoplasmic localization of GFP-only has been shown by several groups, including the Tsien group (Llopis et al., Proc Natl Acad Sci USA. 1998, 95: 6803–6808). In summary, the predominant nuclear staining of GFP-only in our experiments was an effect of the fixation and extraction procedure used, and therefore it does not affect the conclusions we have drawn from the experiments in which GFP-only was used as control.

2) Golgi membrane enrichment

Since the authors did not perform immuno-EM to localize PolD1 precisely, one would hope that the membrane fractionation data would be suggestive in terms of its association with the Golgi apparatus. Unfortunately, Figure 5b does not show any enrichment of Golgi proteins in their prep. There is no loading control described or any cytosolic and membrane markers other than the Golgi being used to evaluate membrane enrichment of their prep. To evaluate the enrichment of PolD1 on Golgi membranes, equal amounts of total proteins from fractions of the PNS, the partially enriched intermediate along fractionation procedures, and the final highly enriched Golgi membranes should be loaded and compared. Furthermore, the blots should be probed for markers of the cytosol, ER, mitochondria and PM to demonstrate the specific association of PolD1 with Golgi membranes.

Response:

We wish to emphasize that PolD1 localization at the Golgi is clearly demonstrated by two sets of findings: (1) immunofluorescence staining of PolD1 at the Golgi, and (2) detection of PolD1 in the Golgi membranes enriched by using equilibrium sucrose-gradient centrifugation (Figure 5b, c). Our immunofluorescence imaging data have clearly shown the presence of PolD1 at the Golgi; it is again disappointing to note that the reviewer has overlooked these imaging data. Our immunostaining of PolD1 did not show any ER, mitochondrial, or plasma membrane staining pattern (Figure 5b). To further strengthen the imaging results, we performed equilibrium sucrose-gradient centrifugation, a fully accepted method in the field to

separate membrane organelles, and found that PolD1 was distributed along the gradient in a pattern almost identical to that of a Golgi marker and the PolD1 distribution showed minimal overlap with that of an ER marker (Figure 5c). Taken together, our results convincingly demonstrate the Golgi association of PolD1.

Immunoelectron microscopy (immuno-EM) is a tool often used for defining the localization of proteins at specific regions or subdomains of the Golgi. To date, no report has been published on the immuno-EM localization of γ -tubulin or any other key component of γ TuRCs at the Golgi, and such immuno-EM data will reveal whether or not γ TuRC-induced microtubule nucleation occurs in specific subdomains on the Golgi membranes. However, in the absence of this knowledge about γ TuRCs, the suggested immuno-EM localization of PolD1 is not likely to yield any meaningful information regarding the control of Golgi-based microtubule nucleation by PolD1. We believe that at the current stage, immuno-EM analysis of PolD1 is unlikely to provide any further insight into the observed PolD1 function.

3) *In vitro* microtubule nucleation assay

The authors now use purified γ TuRCs (without the Cdk5Rap2 TuNA domain) and purified PolD1, including wt, the polymerase-deficient and the exonuclease-deficient PolD1 mutants. All three bind to γ TuRCs by co-IP and inhibit γ TuRC-mediated microtubule nucleation in vitro. As a specificity control, the authors include another Pol δ subunit, PolD2, which does not attenuate γ TuRC-mediated microtubule nucleation. However, all three forms of PolD1 (wt, D515V, S605del) were expressed and purified from mammalian HEK293T cells but the negative control PolD2 was expressed and purified from bacteria. The authors should express and purify this crucial control protein also from HEK293T cells to rule out the possibility that a co-purified unspecific protein or factor (that is absent in bacteria) accounts for the observed activity.

Response:

As suggested by the reviewer, we expressed PolD2 in HEK293T cells and isolated the protein under conditions identical to those used for PolD1 proteins. When tested in the nucleation assay, this PolD2 protein exerted no effect on γ TuRC-induced microtubule nucleation, much like the bacterially expressed PolD2. We have now included these data on PolD2 purified from HEK293T cells (Figure 2c and Supplementary Figure 1c) and have revised the manuscript accordingly.

In addition to the above specific points, the author should address their statement “Golgi-derived microtubules are required for Golgi assembly and maintenance” (in abstract and text) more carefully. In the recently published Ref. 6 (which the corresponding author of this manuscript co-authored), Akhmanova’s group demonstrated that a compact Golgi can be assembled in the absence of both centrosomal and Golgi microtubules. So how are these data reconciled with the phenotypes that authors describe here? The authors should examine Golgi-derived microtubules using a cleaner system where centrosomal microtubules are depleted (such as knockout or centrinone treatment), as the Golgi-derived microtubules are so

closely placed to the centrosomes, which very likely confounds authors' quantitation and interpretation.

Response:

We appreciate the reviewer's comment regarding the statement "Golgi-derived microtubules are required for Golgi assembly and maintenance." To describe the role of Golgi-derived microtubules more precisely than we did before, we have revised the statement as "Golgi-derived microtubules are required for the assembly and maintenance of the proper Golgi structure," and this has been demonstrated by studies from several groups (References 2–7 of this manuscript). In our manuscript, we show that Golgi-derived microtubules play a key role in determining the size and compactness of the Golgi. We further show that the overgrowth of Golgi-derived microtubules increases the size and reduces the compactness of the Golgi and causes defects in Golgi reassembly after nocodazole washout (Figure 6). In the study cited as Reference 6 (Wu et al., *Dev Cell*, 2016, 39: 44–60), the results revealed that in the absence of both centrosomal and Golgi microtubules, an abnormally compact Golgi was assembled; this observation supports our conclusion that Golgi-derived microtubules participate in the control of Golgi size and compactness.

The reviewer suggested that Golgi-derived microtubules should be examined in a system in which centrosomal microtubules are depleted using methods such as knockout or centrinone treatment. However, centrosome inactivation by centrinone treatment has been demonstrated to increase the recruitment of γ -tubulin to the Golgi membranes and thus enhance microtubule growth at the Golgi (Wu et al., *Dev Cell*, 2016, 39: 44–60). We have also shown that centrosome loss caused by laser ablation delocalizes CDK5RAP2, a stimulator of γ TuRCs, from the Golgi (Wang et al., *J Biol Chem.*, 2010, 285: 22658–22665). Therefore, the removal of centrosomes or the inactivation of centrosomal microtubule growth alters the protein composition and the microtubule growth capacity of the Golgi. Consequently, the use of such methods in this study assessing Golgi-derived microtubules would be inappropriate and could yield inaccurate and misleading results.

We would also like to stress that we quantified Golgi-derived microtubules after a short time of regrowth (1.5 and 1 min in Figures 3 & 4, respectively). At these time points, centrosomal microtubules exhibited the pattern of a small microtubule aster, whereas Golgi-derived microtubules were observed as short single microtubules attached to the Golgi. Morphologically, we can clearly discriminate between these two microtubule populations and select Golgi-derived microtubules for quantification. Furthermore, at 1.5 min of regrowth, the amount of Golgi-derived microtubules differed drastically between PolD1-overexpressing and control cells: Golgi-derived microtubules were barely detectable in the PolD1-overexpressing cells, whereas numerous Golgi-derived microtubules were observed in the control cells (Figure 3). Similarly, we can clearly observe the difference in the amount of Golgi-derived microtubules between the PolD1-silenced and control cells (Figure 4). Therefore, our quantification of the Golgi-derived microtubules is highly unlikely to have been affected by contaminating centrosomal microtubules.

Reviewer #3:

The revised manuscript with additional experimental data makes the work much more convincing than that previous version. Novel function of POLD1 is exciting link between microtubules and DNA metabolism.

Response:

We thank the reviewer for acknowledging the significance of the findings presented in this manuscript.

Reviewers' Comments:

Reviewer #1:

Remarks to the Author:

I am satisfied with the revision and strongly recommend publication at this stage.

Reviewer #2:

Remarks to the Author:

the authors have addressed all of our comments and we now recommend publication